# Plant Raf-like kinases regulate the mRNA population upstream of ABA-unresponsive SnRK2 kinases under drought stress

Fumiyuki Soma [1], Fuminori Takahashi [2], Takamasa Suzuki [3], Kazuo Shinozaki [2] & Kazuko Yamaguchi-Shinozaki [1]✉

SNF1-related protein kinases 2 (SnRK2s) are key regulators governing the plant adaptive responses to osmotic stresses, such as drought and high salinity. Subclass III SnRK2s function as central regulators of abscisic acid (ABA) signalling and orchestrate ABA-regulated adaptive responses to osmotic stresses. Seed plants have acquired other types of osmotic stress-activated but ABA-unresponsive subclass I SnRK2s that regulate mRNA decay and promote plant growth under osmotic stresses. In contrast to subclass III SnRK2s, the regulatory mechanisms underlying the rapid activation of subclass I SnRK2s in response to osmotic stress remain elusive. Here, we report that three B4 Raf-like MAP kinase kinase kinases (MAPKKKs) phosphorylate and activate subclass I SnRK2s under osmotic stress. Transcriptome analyses reveal that genes downstream of these MAPKKKs largely overlap with subclass I SnRK2-regulated genes under osmotic stress, which indicates that these MAPKKKs are upstream factors of subclass I SnRK2 and are directly activated by osmotic stress.

[1] Laboratory of Plant Molecular Physiology, Graduate School of Agricultural and Life Sciences, The University of Tokyo, Bunkyo-ku, Tokyo 113-8657, Japan. [2] Gene Discovery Research Group, RIKEN Center for Sustainable Resource Science, Tsukuba, Ibaraki 305-0074, Japan. [3] College of Bioscience and Biotechnology, Chubu University, Kasugai, Aichi 487-8501, Japan. ✉email: akys@mail.ecc.u-tokyo.ac.jp

A biotic stresses, particularly osmotic stresses such as drought and high salinity, are adverse environmental factors for plant growth and crop productivity. The plant hormone abscisic acid (ABA) plays important roles in the adaptive responses to these stresses. The proteins belonging to the SNF1-related protein kinase 2 (SnRK2) family include key regulators of osmotic stress responses, and are classified into three subclasses[1–3]. Among these subclasses, subclass III SnRK2 proteins are strongly activated by ABA and act as essential positive regulators of ABA signalling downstream of ABA receptors, such as pyrabactin-resistance1/PYR1-like/regulatory components of ABA receptors (PYR/PYL/RCARs), under osmotic stress conditions[4–10]. The activated subclass III SnRK2s modulate various ABA responses, including the induction of stress-responsive genes, through activation of the bZIP-type transcription factors ABA-responsive element (ABRE)-binding proteins/ABRE-binding factors (AREB/ABFs)[11–15]. In plants, subclass III SnRK2s are activated in response to ABA and osmotic stress in not only an ABA-dependent but also an ABA-independent manner[3,16]. Seed plants have acquired other types of SnRK2s in addition to subclass III SnRK2s, specifically, subclass I SnRK2s, such as SRK2A/SnRK2.4, SRK2B/SnRK2.10, SRK2G/SnRK2.1 and SRK2H/SnRK2.5, which are rapidly activated by osmotic stress prior to ABA accumulation and, unlike subclass III SnRK2s, are not activated by ABA[3,16].

Phenotypic analyses of Arabidopsis mutants defective in some subclass I SnRK2s have revealed that ABA-unresponsive subclass I SnRK2s are involved in plant growth and survival under osmotic stress conditions[17–19], but the functional analysis of these SnRK2s has been delayed compared with that of subclass III SnRK2s. To elucidate the physiological function of subclass I SnRK2-mediated phosphorylation signalling, subclass I SnRK2-interacting proteins were recently analysed by co-IP coupled with LC-MS/MS analysis[20]. Several components of the mRNA-decapping complex, such as VARICOSE (VCS) and DECAPPING 2 (DCP2)[21,22], have been identified as candidate interacting proteins. Among these components, VCS, which is a mRNA-decapping activator, physically interacts with four functional subclass I SnRK2s but not with subclass III SnRK2s, and is phosphorylated by all four functional subclass I SnRK2s in processing bodies (P-bodies) under osmotic stress conditions. A mutant plant with no functional subclass I SnRK2s and VARICOSE-knockdown plants exhibit growth retardation under osmotic stress conditions. The expression of many stress-responsive genes is similarly dysregulated in these plants, and the mRNA decay of these transcripts is decreased in these plants under osmotic stress conditions. Subclass I-type SnRK2s have been identified in seed plants, but not in lycophytes or mosses. The "ABA-activated subclass III SnRK2s-AREB/ABF" signalling module is known to function as the transcriptional activation system in ABA-responsive gene expression in from mosses to seed plants[23]. To enhance their adaptability under constitutively changing stress conditions[20], seed plants are thought to have acquired a post-transcriptional regulation system mediated by the "ABA-unresponsive subclass I SnRK2s-VARICOSE" signalling module, which promotes marked changes in the mRNA populations under osmotic stress conditions, in addition to the transcriptional activation system during the process of evolution.

Although the mechanisms underlying the activation of ABA-activated subclass III SnRK2s have been clarified in detail, the mechanisms responsible for the rapid activation of ABA-unresponsive subclass I SnRK2s in response to osmotic stress remain largely unknown, despite their pivotal roles in adaptive responses under osmotic stress conditions. Here, we attempt to identify novel interactors of subclass I SnRK2s through co-IP coupled with LC-MS/MS analysis. Three B4 Raf-like MAP kinase kinase kinases (B4 Raf-like MAPKKKs) are identified as upstream factors of the ABA-unresponsive subclass I SnRK2-VCS signalling pathway that are activated by osmotic stress.

## Results

**Identification of subclass I SnRK2-interacting proteins.** To increase the current understanding of ABA-independent osmotic stress signalling, we first attempted to identify ABA-unresponsive subclass I SnRK2-interacting proteins in Arabidopsis. Among the four members of the functional subclass I SnRK2s, we selected SRK2A and SRK2G due to their strong expression in root and shoot tissues[20]. Green fluorescent protein (GFP)-tagged SRK2A (SRK2A-GFP) and SRK2G (SRK2G-GFP) were immunoprecipitated from SRK2A-GFP- and SRK2G-GFP-expressing plants treated with 800 mM mannitol for 10 min, respectively. The immunoprecipitates were subsequently analysed using liquid chromatography–tandem mass spectrometry (LC-MS/MS). Each of these LC-MS/MS analyses, using the SRK2A-GFP- and SRK2G-GFP-expressing plants, was performed twice as biological replicates. We excluded the proteins that were detected in GFP-expressing plants from the analytical results. The LC-MS/MS analyses ultimately identified 1308 and 774 proteins (with confidence > 95%) as candidate SRK2A-GFP- and SRK2G-GFP-interacting proteins, respectively (Fig. 1a). A previous study revealed that the phosphorylation/dephosphorylation of subclass I SnRK2s is involved in their activation[24]. We thus focused on 21 protein kinases and five phosphatases among the 731 candidate proteins detected in both the immunoprecipitate from the SRK2A-GFP-expressing plants and that from the SRK2G-GFP-expressing plants for our investigation of upstream factors of the subclass I SnRK2s (Fig. 1a; Supplementary Data 1). Interestingly, we found that three B4 Raf-like MAPKKKs were included in these 26 candidate proteins (Supplementary Data 1). Previous studies showed that these three Raf-like kinases were phosphorylated during the early stages of the osmotic stress response of plants[25]. We then focused on these three Raf-like kinases RAF18, RAF20 and RAF24.

To further investigate the physical interactions between subclass I SnRK2s and the three Raf-like kinases, we conducted a co-immunoprecipitation (co-IP) assay using untreated or mannitol-treated plants expressing both RAF18-GFP and SRK2A-mCherry, SRK2G-mCherry SRK2D/SnRK2.2-mCherry. RAF18 was used for the assay as a representative of the three Raf-like kinases. We observed that the SRK2A-mCherry and SRK2G-mCherry proteins were coimmunoprecipitated with the RAF18-GFP protein, but not with the SRK2D-mCherry protein in extracts from both untreated and mannitol-treated plants (Fig. 1b). We subsequently performed a split-luciferase complementation (Split-LUC) assay using Nicotiana benthamiana leaves. Luciferase signals were detected when SRK2A-cLUC and RAF18-nLUC, RAF20-nLUC or RAF24-nLUC were transiently expressed, but not when both nLUC and SRK2A-cLUC were expressed (Fig. 1c). We also performed a split-LUC experiment using SRK2G-cLUC and detected luciferase signals when SRK2G-cLUC and RAF18-nLUC, RAF20-nLUC or RAF24-nLUC were expressed (Supplementary Fig. 1). These results suggest that RAF18, RAF20 and RAF24 are novel potential candidate interacting proteins with subclass I SnRK2s.

Because subclass I SnRK2s localise to P-bodies under osmotic stress conditions[20], the three Raf-like kinases might physically interact with subclass I SnRK2s in P-bodies under osmotic stress conditions. We analysed the subcellular localisation of RAF18 and found that RAF18-GFP mainly localised to the cytoplasm after water treatment (control), whereas a portion of RAF18-GFP accumulated in punctate structures in response to mannitol and

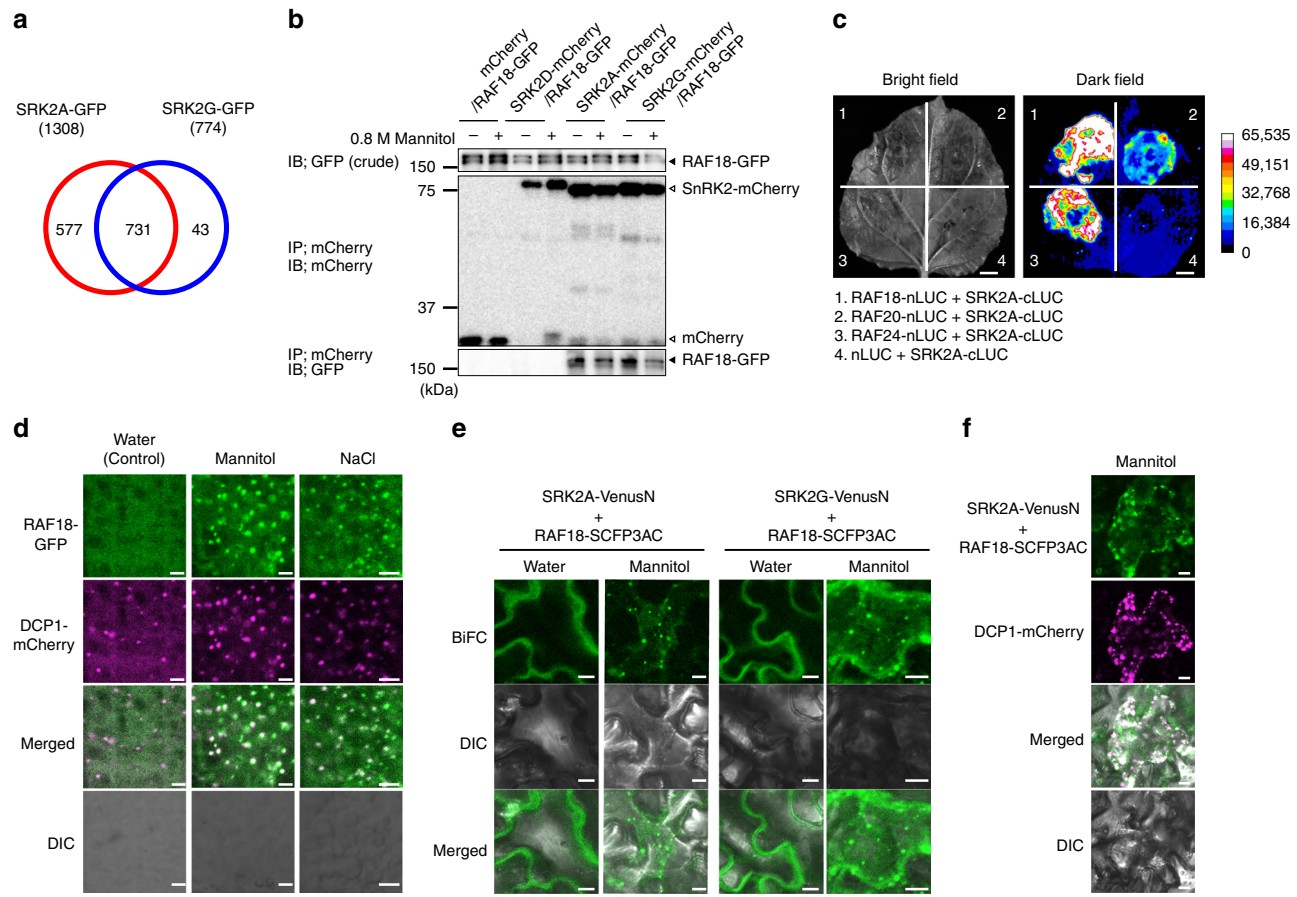

**Fig. 1 The three Raf-like kinases localise and physically interact with subclass I SnRK2s in P-bodies under osmotic stress conditions. a** Venn diagrams showing the numbers of candidate SRK2A-GFP- or SRK2G-GFP-interacting proteins detected in two independent samples under osmotic stress conditions. Each of the analyses was performed twice as biological replicates. **b** The physical interactions of the three Raf-like kinases with subclass I SnRK2s were validated through a co-immunoprecipitation (co-IP) assay. Co-IP was performed using total proteins extracted from transgenic Arabidopsis plants co-expressing RAF18-GFP and mCherry, SRK2A-mCherry, SRK2G-mCherry or SRK2D-mCherry that were treated or not treated with 0.8 M mannitol for 10 min using an anti-mCherry antibody. Immunoblotting was performed with an anti-GFP or anti-mCherry antibody. **c** Split-luciferase assay of the interaction between the three Raf-like kinases and SRK2A in infiltrated *Nicotiana benthamiana* leaves; the bright-field (left) and dark-field (right) results are shown. Scale bars, 1 cm. **d** Confocal images of GFP fluorescence in root cells of transgenic Arabidopsis expressing both RAF18-GFP and DCP1-mCherry and treated with water, 500 mM mannitol or 250 mM NaCl for 30 min. Scale bars, 5 μm. **e** BiFC analyses of the physical interactions between SnRK2s and RAF18 in *N. benthamiana* leaves expressing both SRK2A- or SRK2G-VenusN and RAF18-SCFP3AC and treated with water or 500 mM mannitol for 5 h. Scale bars, 10 μm. **f** Confocal images of fluorescent proteins in *N. benthamiana* leaves expressing SRK2A-VenusN, RAF18-SCFP3AC and DCP1-mCherry. Scale bars, 10 μm.

NaCl treatments (Fig. 1d). Furthermore, punctate RAF18-GFP signals largely overlapped with the signals from the P-body marker DCP1-mCherry (Fig. 1d). These observations suggested that RAF18 localised to P-bodies under osmotic stress conditions. We then validated the physical interaction between subclass I SnRK2s and the three Raf-like kinases at a subcellular level. Bimolecular fluorescence complementation (BiFC) assays showed that the Raf-like kinases interacted with SRK2A and SRK2G in the cytoplasm after water treatment, whereas no detectable interaction between these proteins and MPK6 was found (Fig. 1e; Supplementary Fig. 2). We subsequently performed a BiFC assay under osmotic stress conditions, and detected interactions between RAF18 and SRK2A or SRK2G in punctate structures (Fig. 1e). Furthermore, punctate signals indicating an interaction between RAF18 and SRK2A largely overlapped with DCP1-mCherry signals under osmotic stress conditions (Fig. 1f), which indicated that RAF18 physically interacts with subclass I SnRK2s in P-bodies under osmotic stress conditions.

Previous studies have revealed that subclass I SnRK2s are highly conserved in seed plants, but not in lycophytes or mosses[20]. Therefore, we analysed the phylogenetic relationship

among the Raf-like kinases in various plant species. A molecular phylogenetic analysis revealed that RAF18 (AT1G16270), RAF20 (AT1G79570) and RAF24 (AT2G35050) belong to the group of B4 MAPKKKs[26] (Supplementary Fig. 3). The group of B4 MAPKKKs was widely conserved from mosses to seed plants, whereas RAF18/20/24 have been identified in seed plants, including *Oryza sativa* and *Solanum lycopersicum*, but not in lycophytes and mosses (Supplementary Fig. 3). Based on these results, the thee Raf-like kinases and subclass I SnRK2s might have been specifically acquired by seed plants.

We then evaluated the tissue-specific expression profiles of the three *RAF* genes through the generation of transgenic Arabidopsis plants carrying the promoter of *RAF18*, *RAF20* or *RAF24* fused to the *GUS* gene. GUS activity was widely observed in both the aerial parts and roots of the *RAF18pro:GUS*, *RAF20pro:GUS* and *RAF24pro:GUS* plants, which suggested that the three *RAF* genes are broadly expressed in vegetative tissues (Supplementary Fig. 4a). The expression of *RAF* genes was further validated by quantitative reverse transcription-polymerase chain reaction (quantitative RT-PCR). In addition to the results from the GUS analyses, the three *RAF* genes were widely expressed in both the

aerial parts and roots of the wild-type plants (Supplementary Fig. 4b).

**The three Raf-like kinases activate subclass I SnRK2s.** Importantly, subclass I SnRK2s are rapidly phosphorylated and activated in response to osmotic stress, but the responsible kinases remain unknown. We thus examined whether the three Raf-like kinases could phosphorylate subclass I SnRK2s in vitro. The recombinant glutathione S-transferase (GST)-tagged RAF18 kinase domain (RAF18KD-GST), RAF20 kinase domain (RAF20KD-GST) and RAF24 kinase domain (RAF24KD-GST) enhanced the phosphorylation status of recombinant maltose-binding protein (MBP)-tagged SRK2A (SRK2A-MBP), SRK2B (SRK2B-MBP), SRK2G (SRK2G-MBP) and SRK2H (SRK2H-MBP) in vitro (Fig. 2a; Supplementary Fig. 5a). However, we could not evaluate whether the three Raf-like kinases phosphorylate subclass I SnRK2s or enhance their autophosphorylation activities. Therefore, we generated kinase-inactive forms of subclass I SnRK2s because their autophosphorylation activities made it difficult to assess the possibility of transphosphorylation events between these protein kinases. Lys-33 of SRK2A corresponds to a highly conserved residue that is required for the activity of SnRK2s[27]. Although the mutation of Lys-33 in subclass I SnRK2s to Asn, which produced SRK2A (SRK2AKN), SRK2B (SRK2BKN), SRK2G (SRK2GKN) and SRK2H (SRK2HKN), abolished their autophosphorylation activities (Fig. 2b; Supplementary Fig. 5b), all the SRK2AKN-MBP, SRK2BKN-MBP, SRK2GKN-MBP and SRK2HKN-MBP proteins were phosphorylated by RAF18KD-GST, RAF20KD-GST and RAF24KD-GST (Fig. 2b; Supplementary Fig. 5b). These observations indicated that the three Raf-like kinases can phosphorylate subclass I SnRK2s in vitro. Because VCS was previously shown to be phosphorylated by subclass I SnRK2s[20], in-gel kinase assays of autophosphorylated and RAF20-phosphorylated SRK2A-MBP or SRK2G-MBP were performed using a maltose-binding protein (MBP)-fused VCS fragment (VCSm-MBP) as the substrate to examine the effects of the phosphorylation of subclass I SnRK2s by the Raf-like kinases on these kinase activities. RAF20-phosphorylated SRK2A-MBP and SRK2G-MBP exhibited stronger kinase activities than autophosphorylated SRK2A-MBP and SRK2G-MBP, respectively (Fig. 2c). These data indicate that subclass I SnRK2s are activated through their phosphorylation catalysed by the three Raf-like kinases in vitro.

To determine the sites in subclass I SnRK2s that are phosphorylated by the three RAFs, RAF20-phosphorylated and autophosphorylated SRK2G-MBP as shown in Fig. 2c were analysed by mass spectrometry. We focused on the phosphorylation sites detected in RAF20-phosphorylated SRK2G-MBP, but not autophosphorylated SRK2G-MBP. Nine phosphorylation sites in SRK2G specific to RAF20-phosphorylated SRK2G-MBP were identified (Supplementary Data 2). Among these phosphorylation sites, Ser-154 in SRK2G appears to be an important target residue because it was the first serine in the SxxxS/T motif, which has been shown to be a typical phosphorylation site targeted by various MAPKKKs[28]. Moreover, this Ser-154 residue was conserved among all four subclass I SnRK2s (Supplementary Fig. 6a), and the residue in SRK2B is reportedly phosphorylated in response to osmotic stress[29]. We then performed an analysis of the mutation of Ser-154 in SRK2GKN-MBP to Ala, which resulted in abolishment of the autophosphorylation activity (Fig. 2b; SRK2GKN). The phosphorylation level of SRK2GKN_S154A-MBP by RAF20KD-GST was clearly decreased compared with that of SRK2GKN-MBP (Supplementary Fig. 6b), which suggests that the Ser-154 of SRK2G is one of the direct phosphorylation targets of the three Raf-like kinases.

To elucidate whether subclass I SnRK2s are phosphorylated and activated by the three Raf-like kinases in vivo, we generated a triple mutant of the three *RAF* genes, *raf18/20/24*, by crossing three T-DNA insertion mutants of each gene to one another. A RT-PCR analysis confirmed that the *raf18/20/24* mutant plants exhibited decreased expression of *RAF20* and did not express the other *RAF18* and *RAF24* (Supplementary Fig. 7a, b). The growth phenotypes of the *raf18/20/24* mutant plants grown on agar plates were similar to those of the wild-type plants (Supplementary Fig. 7c, d, e). We then performed an in-gel kinase assay to analyse the kinase activities of subclass I SnRK2s in the *raf* single or multiple mutants. The kinase activity of native subclass I SnRK2s in response to dehydration stress was significantly decreased in the *raf18/20/24* mutant plants, but not in the single- or double-mutant plants (Fig. 2d; Supplementary Fig. 8a). Our in-gel kinase assay also showed that the activation of native subclass I SnRK2s in response to high salinity or mannitol treatment was largely impaired in the *raf18/20/24* mutant plants (Supplementary Fig. 8b). These in-gel kinase assays also found some unknown VCS-phosphorylating kinase activity other than that of SnRK2s (Fig. 2d; Supplementary Fig. 8; asterisk), which was likely due to at least one mitogen-activated protein (MAP) kinase based on the detected molecular weight. The impaired activation of subclass I SnRK2s in the *raf18/20/24* mutant plants was recovered by the expression of *RAF18*, *RAF20* or *RAF24* (Supplementary Fig. 9). Taken together, these results suggest that the three RAFs redundantly and positively regulate the activity of subclass I SnRK2s under osmotic stress conditions in vivo.

We subsequently analysed the phosphorylation status of SRK2B-GFP, which is a subclass I SnRK2, in wild-type or *raf18/20/24* mutant plants because dehydration stress induced clearer electrophoretic mobility shifts of SRK2B-GFP mediated by phosphorylation compared with those obtained with any other subclass I SnRK2s[20]. Clear electrophoretic mobility shifts and activation of SRK2B-GFP were observed in the wild-type plants in response to dehydration stress, but not ABA treatment (Fig. 2e). The shifted bands were confirmed to be phosphorylated forms through treatment with a λ protein phosphatase (λ PP) (Fig. 2e). In contrast, the activation of SRK2D-GFP, which is an ABA-responsive subclass III SnRK2, was observed in response to both dehydration stress and ABA treatment in the wild-type plants (Fig. 2e). The clear electrophoretic mobility shifts and activation of SRK2B-GFP in response to dehydration were largely impaired in the *raf18/20/24* mutant plants (Fig. 2f). In contrast, the activation of SRK2D-GFP in response to both dehydration stress and ABA treatment observed in the *raf18/20/24* mutant plants was similar to that found in the wild-type plants (Fig. 2f). These results suggest that the three Raf-like kinases activate subclass I SnRK2s via their phosphorylation in vivo.

We then performed an in-gel kinase assay to analyse the native kinase activities of subclass III SnRK2s in *raf18/20/24* mutant plants using histone, which can be phosphorylated by both subclass I and III SnRK2s, as the substrate[20]. Several phosphorylated bands were observed in response to ABA treatment in the wild-type plants. However, some of the phosphorylated bands were impaired in the *srk2dei* mutant plants that lack all subclass III SnRK2s (SnRK2D/SnRK2.2, SnRK2E/SnRK2.6 and SnRK2I/SnRK2.3; Fig. 2g), but the phosphorylated bands obtained by subclass III SnRK2s in response to ABA treatment were not affected in the *raf18/20/24* mutant plants (Fig. 2g). We checked the activity of subclass III SnRK2s in response to dehydration in the *raf18/20/24* mutant plants. Several phosphorylated bands were observed in response to dehydration stress in the wild-type plants, but some of these bands were impaired in the *srk2dei* mutant plants (Fig. 2h), and the bands phosphorylated by subclass III SnRK2s were not changed in the *raf18/20/24* mutant

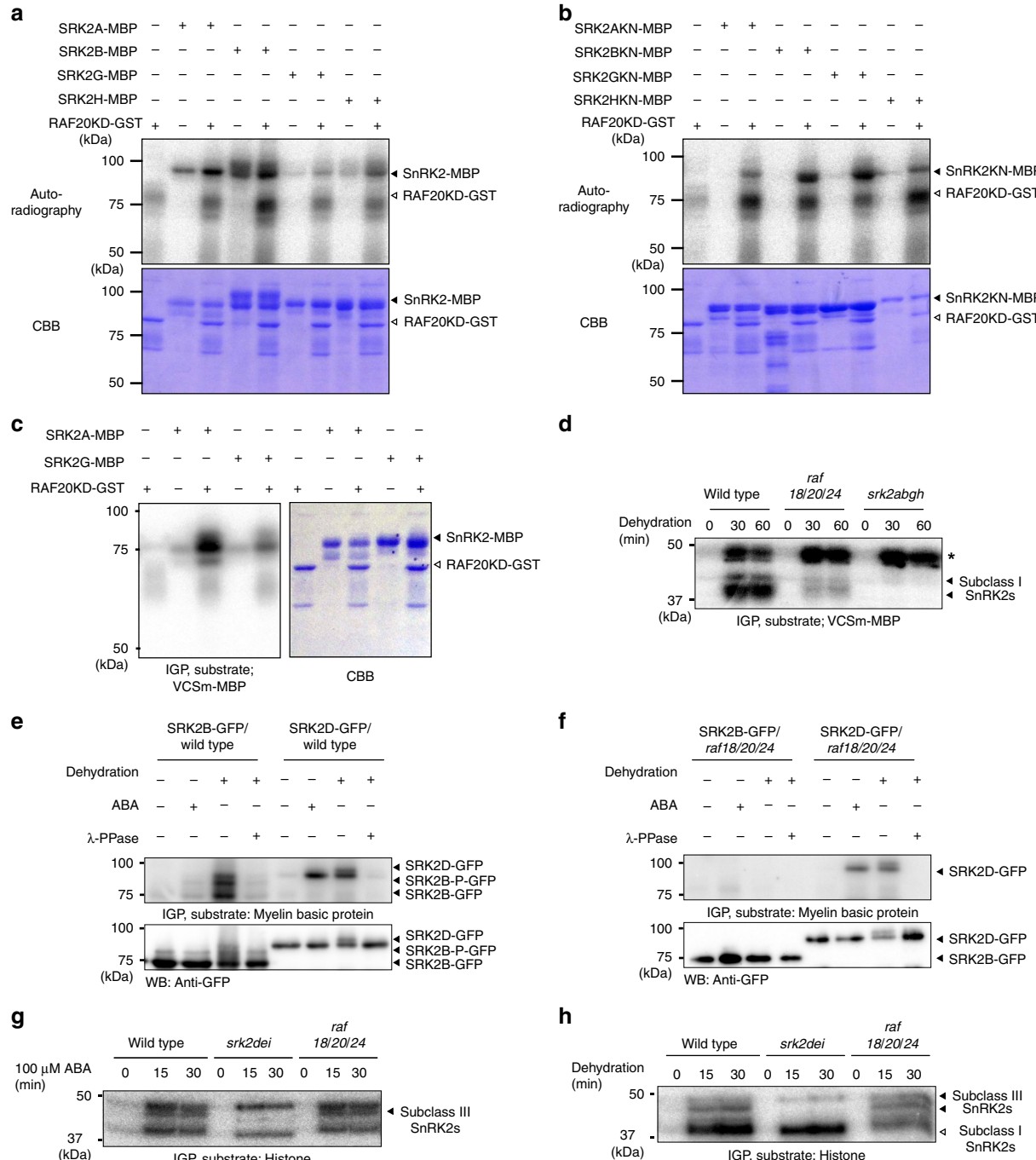

**Fig. 2 The three Raf-like kinases phosphorylate and activate subclass I SnRK2s under osmotic stress conditions. a** Phosphorylation of SRK2A-MBP, SRK2B-MBP, SRK2G-MBP or SRK2H-MBP by RAF20KD-GST in vitro; 500 ng of the recombinant proteins was reacted with g-[$^{32}$P]ATP, electrophoresed on an SDS-polyacrylamide gel and then subjected to Coomassie Brilliant Blue (CBB) staining and autoradiography ($^{32}$P). **b** Phosphorylation of SRK2AKN-MBP, SRK2BKN-MBP, SRK2GKN-MBP or SRK2HKN-MBP by RAF20-GST in vitro. The phosphorylation assays were performed as described in **a**. **c** The phosphorylation of VCSm by SRK2A-MBP or SRK2G-MBP that was in vitro phosphorylated by RAF20KD-GST. In-gel kinase assays (IGP) of recombinant SRK2A-MBP (500 ng) or SRK2G-MBP (500 ng) phosphorylated by RAF20KD-GST (500 ng). The CBB staining and autoradiography results are shown. VCSm-MBP was used as the substrate. **d** IGP assay of VCSm phosphorylation using crude extracts from wild-type, *raf18/20/24* and *srk2abgh* plants in response to drought stress. VCSm-MBP was used as the substrate. An asterisk indicates unknown kinase(s) that can phosphorylate VCSm-MBP. **e**, **f** Electrophoretic mobility and phosphorylation activities of SRK2B-GFP or SRK2D-GFP in wild-type (**e**) and *raf18/20/24* plants (**f**) in response to dehydration stress or ABA treatment. The proteins were purified using an anti-GFP antibody from transgenic Arabidopsis expressing SRK2B-GFP or SRK2D-GFP and subjected to dehydration stress or ABA treatment for 30 min were analysed by immunoblotting (IB) with an anti-GFP or in-gel kinase assay (IGP) using myelin basic protein as the substrate. The electrophoretic mobility of SRK2B-GFP or SRK2D-GFP in response to dehydration after treatment with protein phosphatase (λ-PPase) is also shown. **g**, **h** IGP assay of histone using crude extracts from wild-type, *srk2dei* and *raf18/20/24* plants in response to ABA treatment (**g**) or dehydration stress (**h**).

plants (Fig. 2h). In contrast, the bands phosphorylated by kinases other than subclass III SnRK2s, which appear to be subclass I SnRK2s, were decreased in the *raf18/20/24* mutant plants. These observations indicate that the three Raf-like kinases regulate subclass I SnRK2s under osmotic stress conditions but are not involved in the regulation of subclass III SnRK2s and responses to ABA treatment.

**RAF18/20/24 regulate plant growth under osmotic stress**. To gain insight into the functional relationships between the three Raf-like kinases and subclass I SnRK2s in plants, we examined the growth of the *raf* mutants under osmotic stress-inducing conditions. Our previous studies showed that *srk2abgh* plants lacking all functional *subclass I SnRK2* genes showed growth retardation under osmotic stress conditions[20]. When grown in soil with a limited water supply, the *raf* single- and double-mutant plants showed a similar growth phenotype as that of the wild-type plants (Supplementary Fig. 10). In contrast, under drought stress conditions, the *raf18/20/24* triple-mutant plants showed growth retardation with smaller leaves compared with the wild-type plants, similar to the effects observed in the *srk2abgh* plants (Fig. 3a, b; Supplementary Fig. 10). The enhanced growth retardation detected in the *raf18/20/24* plants was recovered by complementation of the *RAF18*, *RAF20* or *RAF24* genes (Fig. 3a, b; Supplementary Fig. 11). Because the RAF18/20/24-subclass I SnRK2 signalling module regulates plant growth under osmotic stress conditions, the growth of *raf18/20/24* plants in soil might be affected under low-humidity conditions. When grown in soil under 80% humidity (relative humidity; RH, 22 °C), both the *raf18/20/24* and *srk2abgh* plants grew similarly to the wild-type plants (Supplementary Fig. 12). In contrast, when grown under 50% humidity, both mutant plants showed retarded growth compared with the wild-type plants (Supplementary Fig. 12). We subsequently examined the growth of the *raf18/20/24* plants on agar plates under high salinity or osmotic stress conditions. In the presence of 100 mM NaCl or 100 mM mannitol, the *raf* single- and double-mutant plants showed similar growth (Supplementary Fig. 13), whereas the *srk2abgh* and *raf18/20/24* plants showed retarded growth relative to the wild-type plants, as demonstrated by smaller aerial parts and reduced primary root growth (Fig. 3c–e; Supplementary Fig. 13). The enhanced growth retardation detected in the *raf18/20/24* plants was recovered through expression of the *RAF18*, *RAF20* or *RAF24* gene (Fig. 3c–e; Supplementary Fig. 14). These observations suggest that the RAF18/20/24-subclass I SnRK2 signalling module plays important roles in plant growth under osmotic stress conditions. In most of the plant growth experiments, the *raf18/20/24* plants showed slightly retarded growth compared with the *srk2abgh* plants, which implies the existence of other minor targets of the three Raf-like kinases.

**RAF18/20/24 regulates osmotic stress-responsive genes**. Previous studies have revealed that subclass I SnRK2s facilitate marked changes in the population of mRNAs that regulate mRNA decay via phosphorylation of a mRNA-decapping activator, VCS, under osmotic stress conditions[20]. Because the subclass I SnRK2s-VCS module promotes changes in the mRNA population, the three Raf-like kinases might also play roles in regulating the mRNA population under osmotic stress. To examine this hypothesis, we performed transcriptomic analyses of the wild-type, *raf18/20/24* and *srk2abgh* plants following dehydration stress treatment through RNA-sequencing analysis. After dehydration treatment for 5 h, 400 and 989 genes significantly increased and decreased in expression levels, respectively, in the *raf18/20/24* plants compared with those in the wild-type plants

(fold change > 2 or fold change < 0.5, *P* < 0.05) (Fig. 4a, b; Supplementary Data 3). Importantly, many of the up- and down-regulated genes in the *raf18/20/24* under dehydration stress were similarly regulated in the *srk2abgh* plants (Fig. 4a, b). In addition, the expression levels of the up- and downregulated genes in the *raf18/20/24* plants were significantly correlated with their levels in the *srk2abgh* plants, respectively (Fig. 4c). Furthermore, the majority of up- and downregulated genes in the *raf18/20/24* plants were repressed and induced by dehydration, respectively, in the wild-type plants (Fig. 4d). The increased or decreased expression of genes in the *raf18/20/24* plants was further validated by quantitative RT-PCR. mRNA transcripts, such as *CYP75B1*, *MLP423*, *CYSTM8* and *At1g13609*, which were repressed in the wild-type plants by dehydration stress, accumulate to higher levels in both the *raf18/20/24* and *srk2abgh* plants under dehydration stress conditions compared with their levels in the wild-type plants (Fig. 4e). In contrast, various mRNA transcripts, such as *ERF53*, *ERF54*, *At5g25600* and *SIS*, which were induced by dehydration in the wild-type plants, accumulated to lower levels in both *raf18/20/24* and *srk2abgh* plants under dehydration stress compared with their levels in the wild-type plants (Fig. 4f). These results suggest that the three Raf-like kinases regulate the mRNA population under osmotic stress by activating subclass I SnRK2 activity.

**Discussion**
Although subclass I SnRK2s are activated in response to osmotic stress and function in the regulation of mRNA decay, the associated activation mechanism has not been elucidated[20]. Unlike subclass III SnRK2s, these kinases are not activated by ABA and are thought to be directly and promptly activated by osmotic stress prior to ABA accumulation. We thus attempted to elucidate the subclass I SnRK2-mediated signalling system in comparison with the subclass III SnRK2 signalling system. Since subclass I SnRK2s are rapidly activated in response to osmotic stress, we performed LC-MS/MS analyses to identify subclass I SnRK2-interacting proteins in plants that were treated with 800 mM mannitol for a short time (10 min) (Fig. 1a). Considering the possible existence of an activation mechanism that involves the phosphorylation cascade, we focused on 21 protein kinases and five phosphatases which were among the identified candidate interacting proteins (Supplementary Data 1). The data from the previous phosphoproteomic analyses[25] revealed that three B4 Raf-like MAPKKKs, denoted RAF18, RAF20 and RAF24, among the 21 candidate protein kinases were phosphorylated within 10 min in response to osmotic stress. Furthermore, we performed a subcellular localisation analysis, in vitro phosphorylation assays and investigations of plants with multiple mutations in *RAF18*, *RAF20* and *RAF24* and found that these three Raf-like kinases were protein kinases that phosphorylate and activate subclass I SnRK2s in Arabidopsis plants under osmotic stress conditions (Figs. 1 and 2). In addition, we found that the three Raf-like kinases are not activated by ABA and are not involved in the activation of subclass III SnRK2s, which suggests that the RAF18/20/24-subclass I SnRK2-VCS signalling pathway acts independently from the subclass III SnRK2 signalling pathway. Thus, this study highlights the existence of three Raf-like kinases that activate downstream subclass I SnRK2s in response to osmotic stress to form the ABA-independent RAF18/20/24-subclass I SnRK2-VCS signalling cascade (Fig. 4g).

The three Raf-like kinases belong to the family of B4 Raf-like MAPKKKs. Eighty MAPKKKs have been identified in the Arabidopsis genome, and these have been classified into three groups denoted A to C. Among the 80 MAPKKKs, 48 MAPKKKs classified into groups B and C are Raf-like kinases. In addition, the

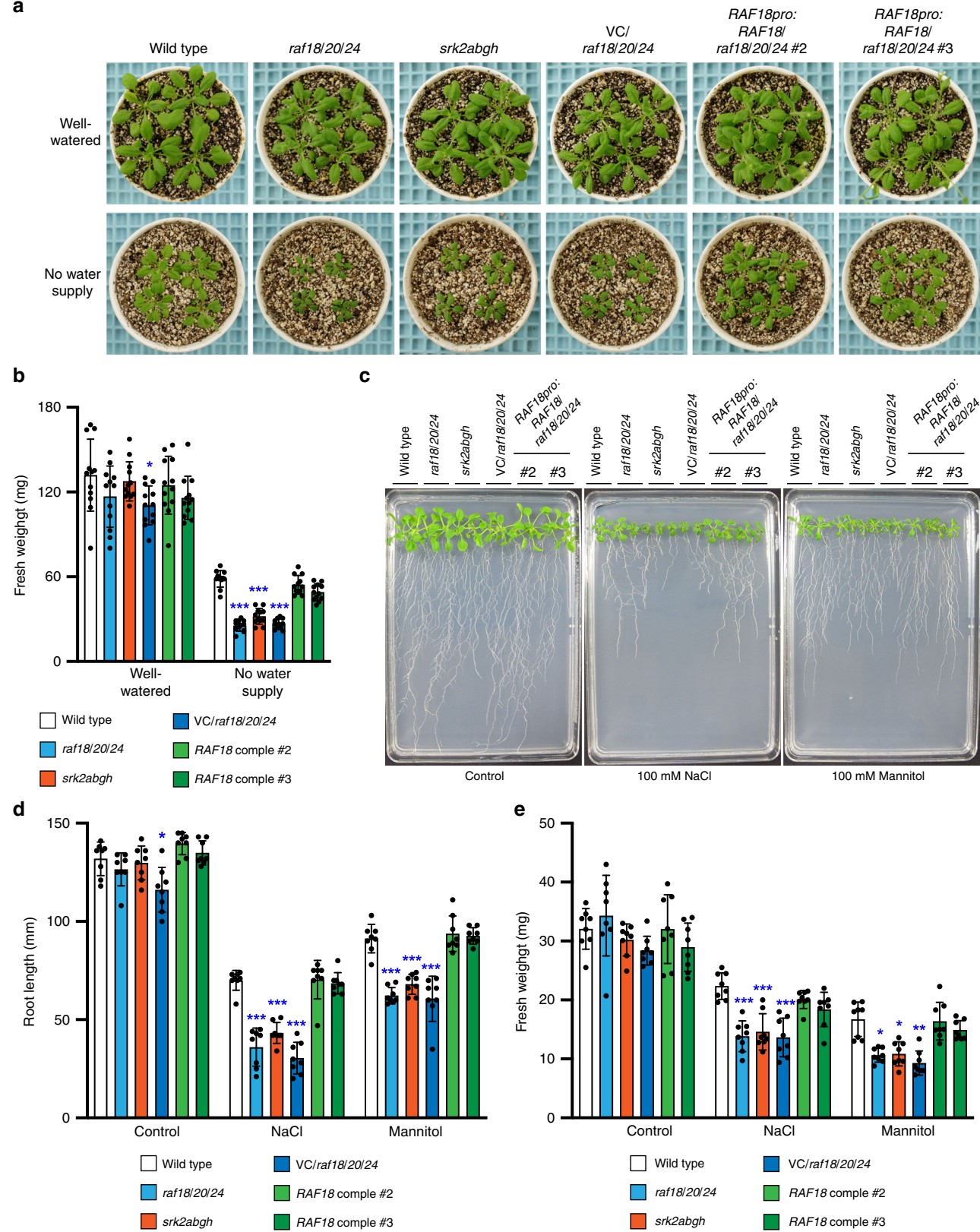

Raf-like MAPKKKs classified into group B are divided into four subgroups denoted B1 to B4[26,30] (Supplementary Fig. 3). The B4 subgroup has seven members in Arabidopsis, and three of these Raf-like kinases share high homology with each other. Three of the other four members of B4 have not yet been characterised, and the last member, HYDRAULIC CONDUCTIVITY OF ROOT 1 (HCR1), is a negative regulator of root water permeability[31]. Although the B4 subgroup has members found in mosses to seed plants, the three Raf-like kinases are evolutionarily divergent from the members found in lycophytes or mosses (Supplementary Fig. 3). Subclass III SnRK2s are highly conserved from mosses to seed plants, but subclass I SnRK2s have not been

**Fig. 3 The *raf18/20/24* mutant displays enhanced growth retardation under osmotic stress conditions. a** Growth phenotypes of wild-type, *raf18/20/24*, *srk2abgh*, VC/*raf18/20/24* (harbouring the vector), and *RAF18*/*raf18/20/24* (harbouring *RAF18pro:RAF18*) plants grown under water-limited conditions. The plants were grown on germination medium (GM) agar plates for 12 days, in soil for an additional 2 days and subsequently without water for 10 days. **b** Fresh weight (mg) of the aerial parts of plants grown as described in **a**. The data are presented as the means ± s.d.s (*n* = 12). An asterisk shows that the indicated mean was significantly different from the mean value of the wild-type plant under the corresponding condition (*P < 0.01, **P < 0.001, ***P < 0.0001, two-way ANOVA followed by Tukey's test). **c** Growth phenotypes of wild-type, *raf18/20/24*, *srk2abgh*, vector/*raf18/20/24* and *RAF18*/*raf18/20/24* plants grown under high salinity or osmotic stress conditions. The photographs show representative phenotypes of 14-day-old seedlings grown vertically on half-strength Murashige–Skoog agar plates containing 100 mM NaCl or 100 mM mannitol. **d**, **e** Primary root growth (**d**) and fresh weight (**e**) of seedlings treated as described in **c**. The data are presented as the means ± s.d.s (*n* = 8). An asterisk shows that the indicated mean is significantly different from the mean value of the wild-type plant under the corresponding condition (*P < 0.01, **P < 0.001, ***P < 0.0001, two-way ANOVA followed by Tukey's test).

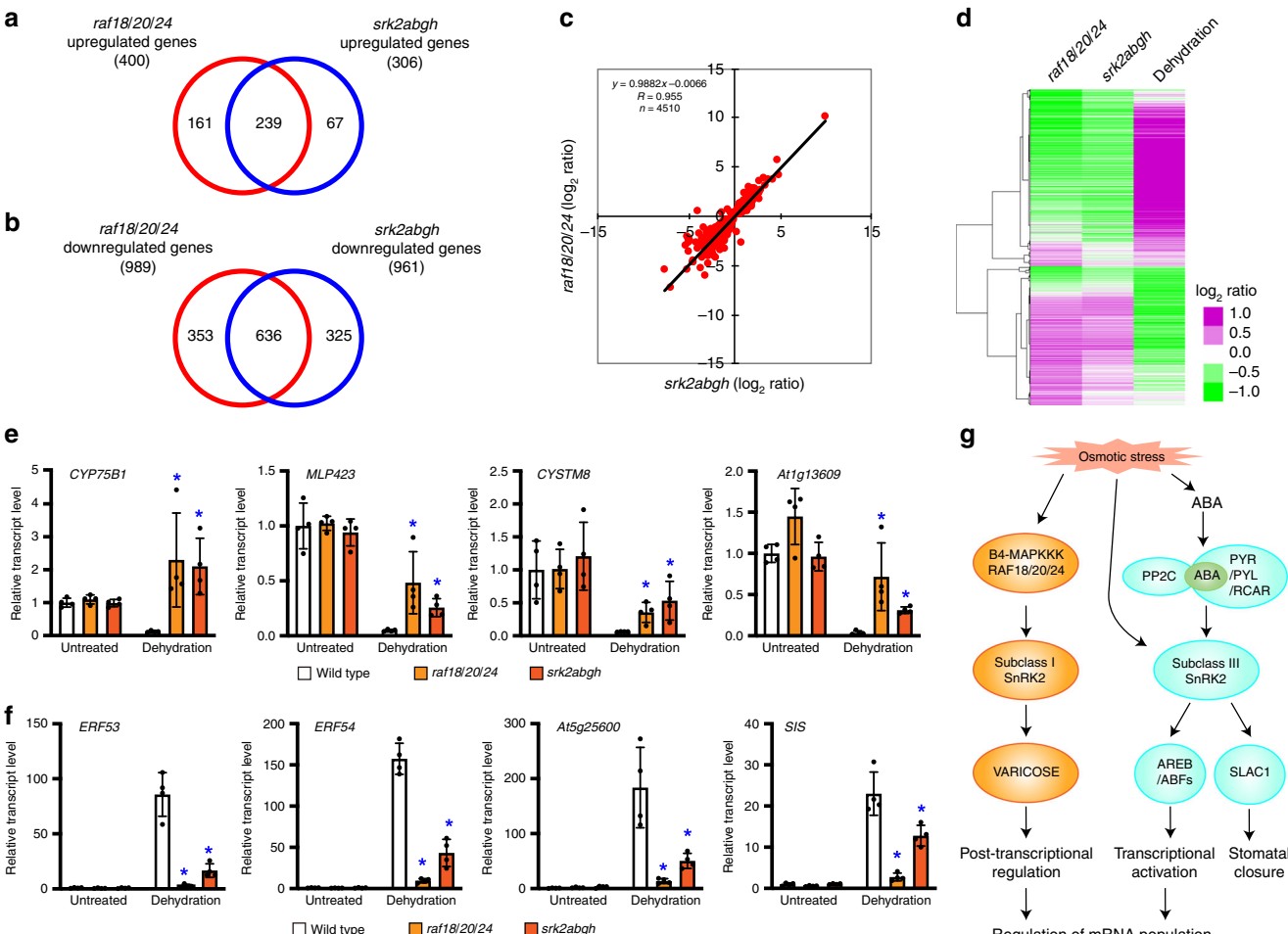

**Fig. 4 Transcriptome analysis of RAF18/20/24- and subclass I SnRK2-regulated genes in response to osmotic stress.** RAF18/20/24-regulated genes largely overlapped with subclass I SnRK2-regulated genes under dehydration stress conditions. **a** Venn diagram showing the upregulated genes (*P* < 0.05, fold change > 2.0) in the *raf18/20/24* and *srk2abgh* mutants after treatment with dehydration for 5 h. **b** Venn diagram showing the downregulated genes (*P* < 0.05, fold change < 2.0) in the *raf18/20/24* and *srk2abgh* mutants after dehydration treatment for 5 h. **c** Analysis of the correlations between the genes downstream of the three RAFs and subclass I SnRK2s. The induction ratios of genes that were significantly changed (*P* < 0.05) in both mutant plants under dehydration are shown in the scatterplots together with the corresponding regression equation and correlation coefficient. **d** Hierarchical clustering analysis of RAF18/20/24-regulated genes. Left column: the expression levels of 6804 genes that were significantly up- or downregulated in the *raf18/20/24* mutant compared with the wild-type expression levels under dehydration conditions. Middle column: the expression levels of the same 6804 genes in the *srk2abgh* mutant relative to the wild-type expression levels under dehydration conditions. Right column: the wild-type expression levels of the 6804 genes under dehydration stress conditions. **e**, **f** Expression levels of upregulated (**e**) or downregulated genes (**f**) in the *raf18/20/24* and *srk2abgh* mutants under untreated or dehydration stress conditions. The expression level in the untreated wild-type plants was defined as 1.0. The values are presented as the means ± s.d.s from four biological replicates. An asterisk shows that the indicated mean was significantly different from the mean value of the wild-type plant under the corresponding condition (*P < 0.05, two-tailed *t* test with Bonferroni correction). **g** Schematic model illustrating that in parallel with transcriptional activation mediated by the "ABA activated subclass III SnRK2-AREB/ABF" signalling pathway, the "RAF18/20/24-subclass I SnRK2-VARICOSE" signalling pathway mediates post-transcriptional regulation to control the mRNA population. Three newly identified RAFs are B4 Raf-like MAPKKKs and transduce osmotic stress by phosphorylating subclass I SnRK2s.

identified in lycophytes and mosses, which is consistent with the finding that members with high homology to the three Raf-like kinase have not been identified in lycophytes or mosses. Subclass I SnRK2s are thought to have been acquired by seed plants to enhance their adaptability to osmotic stress conditions through the control of mRNA accumulation through VCS regulation[20]. The Raf-like kinases appear to be activated by osmotic stress but not ABA, similar to subclass I SnRK2s (Fig. 2). The Raf-like kinases might have evolved with the emergence of the ABA-unresponsive subclass I SnRK2-VCS signalling module in seed plants.

In the moss *Physcomitrella patens*, an ancestral B3-MAPKKK, ABA AND ABIOTIC STRESS-RESPONSIVE RAF-LIKE KINASE (ARK), was previously identified as an upstream factor of ABA-responsive SnRK2s (PpSnRK2s)[32]. ARK activates PpSnRK2s in response to ABA and osmotic stress. All PpSnRK2s encoded in the *P. patens* genome are classified into the subclass III category[33]. A generally accepted mechanism for the activation of subclass III SnRK2 is the following: the binding of ABA to PYR1/PYL/RCAR receptors inhibits group A PP2Cs, which are negative regulators of SnRK2s, and subclass III SnRK2s are thereby activated through autophosphorylation[4,5]. However, it has been reported that the activation of subclass III type PpSnRK2s in response to ABA requires phosphorylation by the upstream ABA-activated ARK protein kinase in *P. patens*[32]. Arabidopsis subclass III SnRK2s are known to be activated through both ABA-dependent and ABA-independent pathways[16]. In Arabidopsis, six members belong to the subgroup of B3 Raf-like MAPKKKs (Supplementary Fig. 3), and three of these members complement the activity of ARK in *P. patens*[32]. Because Arabidopsis B4 Raf-like MAPKKKs specifically activate subclass I SnRK2s in response to osmotic stress, these B3 Raf-like MAPKKKs might activate subclass III SnRK2s in response to ABA and osmotic stress in an ABA-dependent and/or ABA-independent manner in plants. Indeed, while this paper was under review three B3 Raf-like MAPKKKs were reported to be required for ABA and osmotic stress-responsive activation of subclass III SnRK2s[34]. Moreover, Arabidopsis Raf10 and Raf11, which belong to the subgroup of B2 MAPKKKs, are reportedly positive regulators of ABA responses, including the establishment of seed dormancy and glucose/abiotic stress responses, in plants[35]. Although the target proteins of these two kinases have not yet been identified, they might be subclass III SnRK2s because B2 MAPKKKs are more closely related to B3 than B4 MAPKKKs. Further studies using Arabidopsis with various multiple mutations are necessary to clarify the roles of these B2 and B3 Raf-like MAPKKKs in the regulation of the activity of subclass III SnRK2 in plants.

mRNA decapping and subsequent mRNA degradation play important roles in the regulation of the mRNA population to allow adaption to various environmental stress conditions. The activation of mRNA degradation prior to the induction of transcription is considered to result in the activation of stress-induced gene expression and thereby lead to efficient changes in the mRNA population. We previously reported that subclass I SnRK2s play important roles in the regulation of mRNA decay under osmotic stress conditions by phosphorylating the mRNA-decapping activator VCS[20]. In this study, we revealed that the Raf-like kinases positively regulate the expression of drought stress-responsive genes by activating the subclass I SnRK2-VCS signalling module to promote plant growth under osmotic stress conditions. The growth of the *raf18/20/24* and *srk2abgh* plants and the differentially regulated genes in these mutant plants under osmotic stress conditions were very similar (Figs. 3 and 4), which suggests that the main target proteins of the Raf-like kinases are subclass I SnRK2s and that the activation of the Raf-

like kinases is essential for the activation of subclass I SnRK2s under osmotic stress conditions. These results imply that plant growth retardation caused by osmotic stresses, such as drought and salt stress, can potentially be improved by controlling the RAF18/20/24-subclass I SnRK2-VCS signalling pathway.

The next challenge was to identify upstream factors that activate the Raf-like kinases. The factors upstream of a B3 Raf-like MAPKKK, CONSTITUTIVE TRIPLE RESPONSE 1 (CTR1), have been well studied[36,37]. In Arabidopsis, CTR1 negatively regulates ethylene signalling in conjunction with a family of receptors under normal conditions. Upon binding to ethylene, the receptors transmit the signal to the CTR1 protein kinase and thereby inhibit the ability of the kinase to phosphorylate its target protein, ETHYLENE INSENSITIVE 2 (EIN2), which activates downstream ethylene signalling. Ethylene receptors, such as ETHYLENE RESPONSE 1 (ETR1), belong to a family of two-component histidine kinases. The set of Arabidopsis histidine kinases consist of 11 members. In plants, five of these members are ethylene receptors, five of the remaining members [CYTO-KININ INDEPENDENT (CKI) 1, CKI2, ARABIDOPSIS HISTI-DINE KINASE (AHK) 2, AHK3 and AHK4/CYTOKININ RESPONSE 1 (CRE1)] function as cytokinin receptors, and ATHK1 serves as a putative osmosensor[38–41]. Furthermore, complementation experiments with the yeast osmosensor mutant *sln1* and Arabidopsis overexpressor and mutant analyses suggest that the cytokinin receptors AHK2, AHK3, and CRE1/AHK4 have dual functions as osmosensors[40]. Because the sensors upstream of CTR1 are histidine kinases, these candidate osmosensors appear to be factors upstream of the Raf-like kinases. Moreover, upstream factors might be common to subclass III SnRK2s, and the identification of upstream factors of the Raf-like kinases will reveal whether these upstream factors are common or different.

## Methods

**Plant materials and generation of transgenic plants**. *Arabidopsis thaliana* (L.) Heynh ecotype Columbia (Col) was used. The seeds were surface-sterilised and sown on plates with germination medium (GM) agar [4.6 g/L Murashige and Skoog Plant Salt Mixture (FUJIFILM Wako Chemicals), 1 mL/L GAMBORG'S vitamin solution (SIGMA), 30 g/L sucrose, 8.3 g/L Bacto Agar (BD Biosciences) and 0.5 g/L MES, pH 5.7 (KOH)]. After stratification at 4 °C for 3 days in the dark, the plates were incubated in a growth chamber under a 16-h light/8-h dark photoperiod $(60 \pm 10 \, \mu E \, m^{-2} \, s^{-1})$ at 22 °C. For plant growth in soil, 12- to 14-day-old seedlings grown on GM agar plates were transferred into a plastic pot (diameter of 8 cm, height of 6.5 cm) filled with perlite-containing soil (Dio Chemicals) and grown under a 16-h light/8-h dark photoperiod $(60 \pm 10 \, \mu E \, m^{-2} \, s^{-1})$ at 22 °C and 70–85% relative humidity. The T-DNA insertion lines *raf18* (SALK_053373), *raf20* (SALK_069912) and *raf24* (SALK_107170) were obtained from the Arabidopsis Biological Resource Center (ABRC, Columbus, OH, USA) or the European Arabidopsis Stock Centre (NASC, Nottingham, UK). A series of multiple *raf* mutants in the Col ecotype were constructed by genetic crosses and screened via genomic PCR using the recommended primers (ABRC). The *srk2abgh* mutant was constructed by genetic crosses and via genomic PCR[20], and the plants were transformed using *Agrobacterium tumefaciens* strain GV3101 by the floral-dip method[42]. The GV3101 was infiltrated to *Nicotiana benthamiana* plants for transient expression[29,43]. Five-week-old *N. benthamiana* plants were used for *Agrobacterium*-mediated transient expression.

**Co-immunoprecipitation**. Proteins were extracted from approximately 100 3-week-old seedlings grown on GM agar plates and treated or not treated with 0.8 M mannitol for 10 min. Crude proteins were extracted in 10 mL of extraction buffer [20 mM HEPES-KOH (pH 7.6), 100 mM NaCl, 0.1 mM EDTA, 5 mM MgCl₂, 20% (v/v) glycerol and 0.5% (v/v) Triton X-100], and subsequently subjected to two-step centrifugation for the removal of cellular debris. The supernatant was incubated with 120 µL of µMACS Anti-GFP MicroBeads (130-091-125; Miltenyi Biotec) for 30 min at 4 °C. The mixtures were applied to M columns (130-042-801; Miltenyi Biotec) placed onto the MiniMACS Separator. The columns were washed with 10 mL of extraction buffer. Co-immunoprecipitation of mCherry-fused proteins was performed using Anti-RFP mAb-Magnetic Beads (M165-11, MBL) according to the manufacturer's recommended protocol. Transgenic Arabidopsis plants expressing SRK2A-sGFP or SRK2G-sGFP driven by the 35S promoter were generated by introducing pGH-35Spro:SRK2A or pGH-35Spro:SRK2G into wild-type

plants, respectively[20]. The SRK2A-GFP and SnRK2G-GFP protein levels were similar in these transgenic plants.

**Peptide preparation for tandem mass spectrometry analysis**. After electrophoresis, immunoprecipitates were fixed and stained with Coomassie brilliant blue. Then, the immunoprecipitates were digested using Trypsin/Lys-C Mix (V5073; Promega) by the in-gel digestion method[44]. The digested peptides were recovered from the gel pieces after adding 50% (v/v) acetonitrile/5% (v/v) formic acid and subsequently desalted using GL-Tip SDB and GL-Tip GC columns (GL Sciences, Tokyo, Japan). The extracted peptides were dried in a vacuum evaporator, and subsequently dissolved in 80% (v/v) acetonitrile containing 0.1% (v/v) formic acid.

**Mass spectrometric analysis and database searches**. Mass spectrometric analysis was performed using a TripleTOF 5600 instrument (SCIEX) with an Autosampler-2 1D Plus and NanoLC Ultra (Eksigent). Each sample was isolated using a MonoCap C18 High-Resolution 2000 column (2000 mm × 100-μm inside diameter, 2-μm pore size; GL Science, Japan). Four microliters of the sample were concentrated through the analytical column at a flow rate of 500 nL/min for 30 min. The mobile phase comprised of 2% acetonitrile and 0.1% formic acid (A) and 80% acetonitrile and 0.1% formic acid (B). The following linear gradient was used in this analysis: A:B = 98:2 at 0 min to A:B = 60:40 over 300 min, A:B = 10:90 over 20 min and A:B = 98:2 over 40 min. The MS scan range was a mass/charge ratio ($m/z$) of 400 to 1250, and the top 20 precursor ions were selected for subsequent MS/MS scans in the high-sensitivity mode. The MS/MS data were analysed using ProteinPilot 5.0 software (SCIEX) and subsequently annotated using the *A. thaliana* TAIR10 protein database for peptide identification.

**Subcellular localisation and BiFC**. A genomic sequence of *RAF18* was amplified from the genome extracted from Col-0 and cloned into a pGH-GFP vector[11] to produce a pGH-RAF18-sGFP vector. A pGKX-DCP1-mCherry plasmid was constructed by inserting the *DCP1* coding region into pGKX-mCherry[20]. For BiFC analysis, the genomic sequences of *RAF18*, *RAF20* and *RAF24* were inserted into the pSCYCE vector[43], and the coding sequences of *SRK2A* and *SRK2G* were inserted into the pVYNE vector[43]. For fluorescence observations, root or leaf cells were observed under a confocal laser-scanning microscope (LSM5 PASCAL, Carl Zeiss). The primers used for generating the constructs are listed in Supplementary Data 4.

**Split-luciferase complementation assays**. The genomic sequences of *RAF18*, *RAF20* and *RAF24* were inserted into the pCAMBIA-NLuc vector[45], and a coding sequence of *SRK2A* was inserted into the pCAMBIA-CLuc vector[45]. Transient expression in *N. benthamiana* leaves was achieved as above. After infiltration of the luciferin reaction solution [0.1 mM luciferin, 10 mM MgCl₂ and 10 mM MES-KOH (pH 7.5)], the luciferase activity was detected with an Image Quant LAS-4000 biomolecular imager (GE Healthcare), and the data were visualised using Fiji[46]. The primers used for generating the constructs are listed in Supplementary Data 4.

**In-gel kinase assays**. Total proteins were extracted from 12-day-old seedlings that were untreated or subjected to dehydration, 0.8 M mannitol or 0.5 M NaCl treatment for the indicated time periods. To obtain each sample, seedlings from five plants were ground to a powder in liquid nitrogen and homogenised at 4 °C in 150 μL of extraction buffer (50 mM HEPES-KOH [pH 7.5], 5 mM EDTA, 5 mM EGTA, 2 mM DTT, 25 mM NaF, 1 mM Na₃VO₄, 50 mM β-glycerophosphate, 20% (v/v) glycerol and one tablet of complete ULTRA protease inhibitor cocktail tablet per 25 mL). The crude extracts were subsequently centrifuged at 20,000 *g* and 4 °C for 30 min for the removal of cellular debris. The supernatants were collected and subjected to SDS-PAGE. The gel was washed in washing buffer [25 mM Tris-HCl pH 7.5, 5 mM NaF, 0.5 g/L BSA, 0.1 mM Na₃Vo₄, 0.5 mM DTT, 0.1% (v/v) Triton X-100]. The gel was further incubated in renaturation buffer (25 mM Tris-HCl pH 7.5, 5 mM NaF, 0.1 mM Na₃VO₄, 1 mM DTT) at 4 °C overnight. In-gel kinase assays were performed in reaction buffer (25 mM Tris-HCl pH 7.5, 0.1 mM Na₃Vo₄,12 mM MgCl₂, 2 mM DTT, 1 mM EGTA) with 50 μCi [γ-³²P]-ATP for 90 min at room temperture[20,27]. MBP-tagged VCSm[20] recombinant protein was used as the substrate. A dephosphorylation assay was performed using λ-Protein Phosphatase (New England Biolabs) according to the manufacturer's recommended protocol.

**Immunoblot analysis**. The protein extracts were separated via SDS-PAGE and then subjected to immunoblot analysis[27]. Immunoblotting was performed using a monoclonal anti-GFP (1:2,500 dilution, 11814460001; Roche) or a monoclonal anti-mCherry antibody (1:2000 dilution, [1C51]ab125096; Abcam) as the primary antibody and stabilised peroxidase conjugated goat anti-mouse antibody (1:10,000, 32430, Thermo Scientific) as the secondary antibody. Signals were developed using the ECL Select Western Blotting Detection Reagent (GE Healthcare) according to the manufacturer's instructions, and detected using the Image Quant LAS-4000 biomolecular imager (GE Healthcare).

**In vitro kinase assays**. pMAL-c2X-SRK2A, pMAL-c2X-SRK2B, pMAL-c2X-SRK2G and pMAL-c2X-SRK2H were constructed by inserting the SRK2A, SRK2B, SRK2G or SRK2H cording region into pMAL-c2X (New England Biolabs)[20]. To generate kinase-negative forms of the SRK2A, SRK2B, SRK2G and SRK2H fragments, site-directed mutagenesis was performed by inverse PCR[27,47], which resulted in the production of pMAL-c2X-SRK2AKN(K33N), pMAL-c2X-SRK2BKN(K33N), pMAL-c2X-SRK2GKN(K33N) and pMAL-c2X-SRK2HKN (K33N), respectively. Fragments of the kinase domains of *RAF18* (2499-3402 bp), *RAF20* (2802-3681 bp) and *RAF24* (2832-3708 bp) were amplified from cDNAs derived from Col-0 plants, and these fragments were cloned into pGEX-4T2 (GE Healthcare Life Sciences) to produce pGEX-4T2-RAF18KD, pGEX-4T2-RAF20KD and pGEX-4T2-RAF24KD, respectively. The resulting plasmids were transformed into *Escherichia coli* strain BL21-Gold (DE3; Agilent Technologies). The recombinant proteins were expressed in bacteria and affinity purified according to the manufacturer's instructions. For in vitro phosphorylation assays, the purified proteins were incubated in a total volume of 15 μL of reaction buffer (50 mM Tris-HCl pH 7.5, 100 mM NaCl, 10 mM MnSO₄, 10 mM MnCl₂, 0.5 mM CaCl₂, 2 mM DTT, 10 μCi[γ-³²P]ATP) for 60 min at 30 °C. The protein samples were then separated by SDS-PAGE, and the gel was subsequently dried and exposed to a Fujifilm imaging plate (BAS-MS; GE Healthcare Life Sciences) for 1d. A radioactive image was visualised by FLA-5000 Phosphor Imager (Fujifilm). The protein level was analysed by Coomassie Brilliant Blue (CBB) staining. The primers used for generating the constructs and site-directed mutagenesis are listed in Supplementary Data 4.

**Physiological assays**. For the assessment of growth under drought stress conditions, seeds were sown on GM agar plates, stratified at 4 °C for 3 days in the dark, and subsequently grown under a 16-h light/8-h dark photoperiod (60 ± 10 μE m⁻² s⁻¹) at 22 °C for 12 days. The seedlings were transferred from GM agar plates to soil, grown in soil for an additional 2 days, and then treated with or without water for 10 days under 80% relative humidity. For the assessment of NaCl or mannitol sensitivity at the seedling stage, seeds were sown on GM agar plates solidified with 1.2% (w/v) Bacto agar (BD Biosciences), stratified at 4 °C for 3 days in the dark, and subsequently grown vertically under a 16-h light/8-h dark photoperiod (60 ± 10 μE m⁻² s⁻¹) at 22 °C. Four-day-old seedlings were transferred to fresh agar plates containing half-strength MS medium, 1% (w/v) sucrose and 0.5 g/L of MES-KOH (pH 5.7) supplemented with 100 mM NaCl or 100 mM mannitol and solidified with 1.2% Bacto Agar and subsequently grown vertically for an additional 11 days. Photographs were then obtained, and the relative primary root growth of each plant was determined. The humidity in the Plant Growth Chamber (Nippon Medical & Chemical Instruments Co., LTD) was controlled according to the manufacturer's recommended protocol.

**RNA extraction**. The total RNA was prepared from 12-day-old seedlings grown on GM agar plates. Five plants were pooled to obtain a single RNA sample. The extraction of the total RNA from the plants was conducted with RNAiso Plus (Takara Bio). The extracted RNA was subjected to RNA sequencing and RT-qPCR analyses.

**RNA sequencing and data analysis**. Three independent RNA samples were used as biological replicates in each RNA-sequencing experiment. The total RNA (1 μg) was used to prepare each library. mRNA purification was performed using a NEBNext Poly(A) mRNA Magnetic Isolation Module (New England Biolabs). cDNA libraries were constructed using the NEBNext Ultra II RNA Library Kit for Illumina (New England Biolabs) and NEBNext Multiple Oligos for Illumina (New England Biolabs), and the libraries were sequenced by NextSeq 500 (Illumina). The single ends of cDNA libraries were sequenced for 86 nt using the illumine Genome Analyzer IIx[48]. The reads were mapped to the Arabidopsis reference (TAIR10). A heatmap was generated using Gene Cluster 3.0[49] and Java TreeView[50] to visualise the clustering results.

**Quantitative RT-PCR**. Four independent RNA samples were used as biological replicates in each quantitative RT-PCR experiment. First-strand cDNA was synthesised from 1 μg of the total RNA using a High-Capacity cDNA Reverse Transcription Kit (Thermo Fisher Scientific, Waltham, MA, USA) with random hexamer primers according to the manufacturer's instructions. Quantitative RT-PCR was performed using QuantStudio 3 Real-time PCR (Applied Biosystems), and Power SYBR Green PCR Master Mix (Applied Biosystems) was used for amplification. Arabidopsis 18S rRNA was used as an internal control in the quantitative RT-PCR assay. The primers used for real-time PCR are listed in Supplementary Data 4.

**Accession numbers**. Sequences of genes described in this article can be found in The Arabidopsis Information Resource (http://www.arabidopsis.org/) under the following accession numbers: SRK2A/SnRK2.4 (At1g10940), SRK2G/SnRK2.1 (At5g08590), SRK2B/SnRK2.10 (At1g60940), SRK2H/SnRK2.5 (At5g63650), SRK2D/SnRK2.2 (At3g50500), VCS/VARICOSE (At3g13300), DCP1 (At1g08370), DCP2 (At5g13570), RAF18 (At1g16270), RAF20 (At1g79570), RAF24

(At2g35050), CYP75B1 (At5g07990), MLP423 (At1g24020), CYSTM8 (At3g22235), ERF53 (At2g20880), ERF54 (At4g28140) and SIS (At5g02020).

**Reporting summary**. Further information on research design is available in the Nature Research Reporting Summary linked to this article.

## Data availability

The LC-MS/MS data of SnRK2-interacting proteins have been deposited to the ProteomeXchange Consortium via the PRIDE[51] partner repository with the dataset identifier PXD017371. The RNA-sequencing data were deposited into the DNA Data Bank of Japan under accession number DRA008643. The source data underlying Figs. 1b, 2a–h, 3b, d, e, and 4e, f and Supplementary Figs 4b, 5, 6b, 7b, d–e, 8a–b, 9a–b, 10b, 11b, 12b, 13b, 13b–c and 14b are provided as a Source Data file.

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

## Acknowledgements

The authors would like to thank Yuriko Tanaka, Saho Mizukado, Tomomi Shinagawa and Ayumi Furuta for the excellent technical assistance provided and Etsuko Toma for the skilled editorial assistance provided. This work was financially supported by Grants-in-Aid for JSPS Fellows (No. 18J13854 to F. Soma), Scientific Research on Innovative Areas (No. 15H05960 to K.Y.-S.), and Scientific Research (A) (No. 18H03996 to K.Y.-S.) from the Ministry of Education, Culture, Sports, Science, and Technology of Japan and for Scientific Research from the Mitsubishi Foundation.

## Author contributions

F.S. and K.Y.-S. designed the research. F.S. performed most of the experiments. F.T. performed the mass spectrometric analysis and database searches. T.S. performed the RNA sequencing and data analyses. F.S., K.S. and K.Y.-S. wrote the article. All authors discussed the results and commented on the paper.

## Competing interests

The authors declare no competing interests.
