## [Peer Review File · Nature Communications]

Reviewers' comments:

Reviewer #1 (Remarks to the Author):

Soma et al. report their identification of three B4 Raf-like MAPKKK as upstream kinases for subclass I SnRK2s that activate these SnRK2s in response to osmotic stress. Subclass I SnRK2s have been known for many years as crucial mediators of fast plant responses to osmotic stress, but the mechanism of their activation has been unknown. This is an important advance and the experiments have been competently and convincingly performed. I support publication of the manuscript in Nature Communications.

Minor points:

1. The authors cite 3 papers that reported the first structures of ABA receptors. However, 5 papers were published on these structures simultaneously, and ironically the 2 most complete and comprehensive papers are not cited.
2. The numerical nomenclature for SnRK2s (e.g., SnRK2.1) is more common than the letter-based one (e.g., SRK2G). Please provide both nomenclatures at least once not only for the 4 functional subclass I SnRK2s, but for all SnRK2s mentioned in the manuscript.
3. Please indicate which B4 Raf-like MAPKKK corresponds to which OSRK at the time you introduce OSRK. That can easily be done just by referring to Fig. S2.
4. Fig. 2d: Can you comment on the top band in the blot. If you think this indicates an unknown VCS-phosphorylating kinase, please mark the band with an asterisk on the side and state so.
5. Fig. 2d: The authors convincingly show that OSRK1/2/3 bind subclass I SnRK2s, are required for their full activation, and can phosphorylate them using reconstituted protein domains. It is still possible that OSRKs affect SnRK2 activity in addition indirectly by changing their expression/protein levels. This can be tested by a western blot of the crude extracts from wildtype and *osrk1/2/3* plants probed for subclass I SnRK2, which would further improve the manuscript.
6. Page 9, last sentence: "In contrast to *srk2abgh* plants,..." That statement seems to contradict the data in Fig. 3a,b. Please state correctly.
7. Through all experiments, the *osrk1/2/3* phenotype is more severe than the *srk2abgh* phenotype, suggesting that OSRK1/2/3 have functions beyond phosphorylation and activation of SRKA/B/G/H. It would be useful to briefly discuss this point in the discussion.

Reviewer #2 (Remarks to the Author):

Soma et al describe the discovery of a new set of kinases, named OSRK1-3 (MAPKKKs), that function upstream of subclass I SnRK2s in ABA-independent signaling. As such, the authors have provided both genetic evidence for these kinases as novel modulators of drought/osmotic stress responses and biochemical evidence placing these kinases in the previously described SnRK2ABGH pathway. They further provide molecular evidence supporting their model showing an overlapping set of changes in transcript accumulation in the *osrk123* and *snrk2abgh* higher order mutants. Thus, I feel that the present manuscript makes a significant advancement to our understanding on how plants respond to water stress.

Other than mostly editorial points raised below, my only experimental criticism relates to Figure 2f. The non-specific band in the in-gel kinase assay is problematic as it co-migrates where SnRK2B should be. Even though the lower immunoblot supports the overall conclusion regarding the gel shift being OSRK1/2/3-dependent, it adds a bit of confusion as to whether the phosphorylation is really linked to the kinase activity of SnRK2B. It would be preferable if the authors could repeat this experiment to eliminate this "non-specific" band (as was cleanly shown in Fig 2d).

Editorial Points:

1. In the description of the co-IP experiments (lines 105-111), the authors refer to the more than 1000 proteins identified as 'candidate interactors of SRK2A-GFP'. First, I believe that nobody believes that >1000 proteins really interact with the kinases and are likely to be mostly non-specific precipitating proteins. Second, it does not appear that a proper control was used to subtract these non-specific proteins from the list (i.e. a blank IP from a non-transformed line). As the end result of the manuscript does not rely on this data other than as an entry point to understand how these kinases were selected for further study, I do not think the list of proteins (or the quantitation) is a major issue. However, lacking the proper controls, the authors should avoid referring to this data as revealing "SnRK2-interacting proteins" as this is quite misleading and not supported by the presented data.
2. In line 188 when referring to MBP-fused VCS fragment in the in-gel kinase: I think it is asking quite a bit of a casual reader to remember from the brief mention in the Introduction that Varicose (VCS) was previously shown to be phosphorylated by subclass I SnRK2s. Bringing in that information to inform the experimental design would be extremely helpful.
3. Although it isn't necessary for this paper for the authors to know the identity of the higher MW non-specific kinase activity in Fig 2d (and following), I don't think that it can go without mentioning that there is another dehydration-related kinase activity present that is OSRK1/2/3-independent. It is so obvious that it needs to be at least mentioned.
4. Line 247: I believe the authors meant to say "Similar to the srk2abgh plants" rather than "In contrast to ..."
5. Lines 258-261: This sentence needs to be rewritten as there are a number of errors in both construction and spelling.

Reviewer #3 (Remarks to the Author):

The article by Soma et al. titled Plant Raf-like kinases regulate the mRNA population upstream of ABA unresponsive SnRK2 kinases under drought stress describes the role of subclass I SnRK2s in osmotic stress in particular their interaction and potential phosphorylation by three B4 Raf like MAPKKKs. They employed biochemical analysis and phenotypic assessments on mutants to derive the functional significance of the activation via phosphorylation of subclass I SnRK2s in plant growth under osmotic stress. Overall, their work highlights an important novel contribution on the role of subclass SnRK2s and the Raf like MAPKKKs in osmotic stress and plant growth promotion, which is of interest in wider field in plant biology and biochemistry.

Here I highlight comments and concerns towards improving this work for publication in particular the need for further evidence on target phosphorylation sites that are only targeted by OSRKs.

Major revisions

With the given data in particular supplementary data on IP, it is highly unlikely that an independent research can reproduce the work without the data evidence from this research. This data needs to be provided.

Results

Line 105: By two independent LC-MS/MS runs, do the authors imply technical replicates or biological replicates. It is not clear. If it is technical replicates, I suggest the authors perform biological replicates in addition.

In addition, there is no data or mention about important co-IP controls such as co-IP from a control

GFP expressing plant line or just GFP IP on the wild type plants and how they identify false positive? How did the authors quantify the candidates co-IP proteins, in particular those that the authors selected. Were they expressed quantitatively in the same manner between the SRK2A and SnRK2G. Line 124: For the luciferase assay, did the authors perform this experiment with SRK2G? If not how could they conclude that OSRK1-3 are potent candidate novel interactors of SnRK2s.

Paragraph starting on line 165: Although the data suggest that the three OSRKs positively regulate the activity of the subclass I SnRK2s under osmotic stress, the authors show an indirect potential that OSRKs may phosphorylate SnRK2s and that this phosphorylation is effective for SnRK to catalyse VCS compared to the autophosphorylated form of SnRK2, I think it is important to determine the exact target residues for OSRK phosphorylation that are not subject to autophosphorylation. As it stands, it leaves a hollow in the study that greatly adds value to the role of OSRK in interacting and phosphorylating SRK2 - if its not just facilitating the autophosphorylation of SnRK2s. After the identification of the phosphosites by mass spectrometry authors should correlate with the bioinformatic predictions to unequivocally prove that the sites obtained are indeed targets of OSRKs. Line 319: Is this statement supported by data: Thus, we concluded that the three OSRKs are quickly activated by phosphorylation?

Minor revisions

Define terms like ARK, EIN2 and ETR1 and the rest of the abbreviations not defined?

Check your abbreviations and make sure they are used at least once. For those not, could you remove them

Number of biological and/or technical replicates used need to be stated for example for the co-IP and for transcriptomics.

Supplementary table 1 should also contain all the co-IP proteins identified showing which ones are detected in both replicates and which ones are common between the SRK2A and 2G-GFPs

1. Line 27: change 'osmotic stress to abiotic stress. This sentence is a generalized statement.

2. Line 40: Rephrase as follows- Abiotic stresses such as drought and high salinity and consequently osmotic stress are

Line 101-102: Are these over expressing lines? Its not clear in both the methods and the results sections.

Line 118-119: Did the authors perform the same experiment with OSRK2 and 3? Its not clear.

Authors need to clearly define their terms, they are using dehydration and osmotic stress interchangeably and in some cases for example Line 240 which is heading for water-deficit experiment, they should clearly explain the link to osmotic stress bearing in mind that one is upstream of the other.

Line 242: under osmotic stress conditions. This could rather be rephrased to..... under osmotic stress inducing conditions.

Line 247: Can the authors clearly explain what they mean by poorer growth- how does one measure poorer growth? I suggest that the authors explain by means of phenotypic traits

Line 248: term drought stress- is this for water deficit? Authors need to maintain wording particularly when referring to stresses or at least indicate that x and y are the same.

Line 249: by using the term "expressing" do the authors mean complementation

Line 260: replace poorer with reduced or retarded

Line 275: do the authors mean osmotic stress?

Line 276: deleted showed and replace reduced by decreased in

Line 280: Now it seems like the authors did a drought stress treatment for srk2abgh plants and dehydration for the osrk1/2/3 plants. Correct all your terminology in the paper and be consistent.

Line 298: Do the authors indicate that SnRK2s are functionally activated in their kinetic role or they are referring to gene induction and functional activation? its not quite clear!

Line 301- replace quickly with promptly
Line 303: Replace As by Since
Line 305-306: Although the authors indicate their hope of elucidating the specific subclass 1 SnRK2s interactors, did they attempt to measure the level of ABA at this 10 mins time point?
Line 308: insert "which were " before among...
Line 309: replace 'of the previous' by from the previous
Line 309-311: rephrase the sentence
Line 324: correct MAPKKs to MAPKKKs
Line 330: By separation do the authors mean evolutionary divergent or something else. This needs clarification.
Line 335: replace 'to be' with 'to have been'
Line 336: delete 'levels'
Line 337: replace 'but not activated by ABA' either by ' but not ABA' or 'and not ABA'
Line 351: Six members of which subclass of SnRK2s- be clear
Line 373: replace 'the gene expression that was upregulated and downregulated in these....' by 'the differentially regulated genes in these.... '
Line 378: delete 'the'
Line 403: State the composition of the germination medium
Line 434: Replace "... were previously used..." by "were used as previously described"
Line 423: Can the authors include a brief description of the immunoblot analysis technique used.
Line 448: Add 'of' in the sentence The mobile phase....
Line 450: the A:B = 60:40 in 300 min, Is this correct? How long is the overall run time for each sample?
Line 510-12: The sentence is not clear on how the plants were grown, for how long each in the GM agar and in the soil prior to water withdrawal.

Figures:

1. state the number of reps in Figure 1
2. Figure 1b does not show the IP controls e.g. GFP or mCherry IP for background. This should also be explained in the co-IP and mass spectrometry data analysis in the methods and results sections
3. Figure d-f correct spelling for merged from marged

Responses to reviewers

Reviewers' comments:

Reviewer #1 (Remarks to the Author):

>>> COMMENT 1

Soma et al. report their identification of three B4 Raf-like MAPKKK as upstream kinases for subclass I SnRK2s that activate these SnRK2s in response to osmotic stress. Subclass I SnRK2s have been known for many years as crucial mediators of fast plant responses to osmotic stress, but the mechanism of their activation has been unknown. This is an important advance and the experiments have been competently and convincingly performed. I support publication of the manuscript in Nature Communications.

>>>ANSWER 1

Thank you very much for your kind comments. We have revised our manuscript according to your valuable suggestions as described below.

>>> COMMENT 2

Minor points:

1. The authors cite 3 papers that reported the first structures of ABA receptors. However, 5 papers were published on these structures simultaneously, and ironically the 2 most complete and comprehensive papers are not cited.

>>>ANSWER 2

In response to this helpful comment, we have cited the two indicated papers (L48-49 and 911-915).

>>> COMMENT 3

2. The numerical nomenclature for SnRK2s (e.g., SnRK2.1) is more common than the letter-based one (e.g., SRK2G). Please provide both nomenclatures at least once not only for the 4 functional subclass I SnRK2s, but for all SnRK2s mentioned in the manuscript.

>>>ANSWER 3

In response to this comment, we have provided both nomenclatures at least once for all SnRK2s described in the manuscript (L124 and 263-264).

>>> COMMENT 4

3. Please indicate which B4 Raf-like MAPKKK corresponds to which OSRK at the time you introduce OSRK. That can easily be done just by referring to Fig. S2.

>>>ANSWER 4

According to this comment, we have indicated the B4 Raf-like MAPKKK that corresponds to each OSRK when the latter are introduced (L158-159).

>>> COMMENT 5

4. Fig. 2d: Can you comment on the top band in the blot. If you think this indicates an unknown VCS-phosphorylating kinase, please mark the band with an asterisk on the side and state so.

>>>ANSWER 5

We believe that the top band in the blot indicates an unknown VCS-phosphorylating kinase(s) that is likely a mitogen-activated protein (MAP) kinase(s) based on the molecular weight. In response to this comment, we have marked the band with an asterisk and provide an explanation in the legend for Figure 2. We also provide a description of this kinase activity in the Results section (L234-237).

>>> COMMENT 6

5. Fig. 2d: The authors convincingly show that OSRK1/2/3 bind subclass I SnRK2s, are required for their full activation, and can phosphorylate them using reconstituted protein domains. It is still possible that OSRKs affect SnRK2 activity in addition indirectly by changing their expression/protein levels. This can be tested by a western blot of the crude extracts from wildtype and *osrk1/2/3* plants probed for subclass I SnRK2, which would further improve the manuscript.

>>>ANSWER 6

Unfortunately, we were unable to obtain an antibody against subclass I SnRK2s that is suitable for this experiment, and we thus could not perform a western blot analysis using crude extracts from the wild-type and *osrk1/2/3* mutant plants. However, using the GFP antibody, we indicated that the levels of the SRK2B-GFP protein were stable and did not change significantly even in the *osrk1/2/3* mutant, as shown in Figures 2e and 2d. We have added the following sentence in L256-257 in the manuscript: “These results suggest that OSRKs activate subclass I SnRK2s via their phosphorylation *in vivo*.”

>>> COMMENT 7

6. Page 9, last sentence: “In contrast to *srk2abgh* plants,...”. That statement seems to contradict the data in Fig. 3a,b. Please state correctly.

>>>ANSWER 7

In response to this helpful comment, COMMENT 13 provided by Reviewer #2 and COMMENT 30 provided by Reviewer #3, we have revised the sentence to accurately present the data (L283-286).

>>> COMMENT 8

7. Through all experiments, the *osrk1/2/3* phenotype is more severe than the *srk2abgh* phenotype, suggesting that OSRK1/2/3 have functions beyond phosphorylation and activation of SRKA/B/G/H. It would be useful to briefly discuss this point in the discussion.

>>>ANSWER 8

In response to this comment, we briefly discuss the indicated point (L304-306).

Reviewer #2 (Remarks to the Author):

>>> COMMENT 9

Soma et al describe the discovery of a new set of kinases, named OSRK1-3 (MAPKKKs), that function upstream of subclass I SnRK2s in ABA-independent signaling. As such, the authors have provided both genetic evidence for these kinases as novel modulators of drought/osmotic stress responses and biochemical evidence placing these kinases in the previously described SnRK2ABGH pathway. They further provide molecular evidence supporting their model showing an overlapping set of changes in transcript accumulation in the *osrk123* and *snrk2abgh* higher order mutants. Thus, I feel that the present manuscript makes a significant advancement to our understanding on how plants respond to water stress. Other than mostly editorial points raised below, my only experimental criticism relates to Figure 2f. The non-specific band in the in-gel kinase assay is problematic as it co-migrates where SnRK2B should be. Even though the lower immunoblot supports the overall conclusion regarding the gel shift being OSRK1/2/3-dependent, it adds a bit of confusion as to whether the phosphorylation is really linked to the kinase activity of SnRK2B. It would be preferable if the authors could repeat this experiment to eliminate this “non-specific” band (as was cleanly shown in Fig 2d).

>>>ANSWER 9

Thank you very much for your valuable comments on our manuscript.

In response to this comment, we performed an in-gel kinase experiment using the GFP antibody to eliminate the “nonspecific” band and to show that phosphorylation is linked to the kinase activity of SRK2B. We indicate the results from this experiment in the new Figure 2f.

>>> COMMENT 10

Editorial Points:

1. In the description of the co-IP experiments (lines 105-111), the authors refer to the more than 1000 proteins identified as ‘candidate interactors of SRK2A-GFP’. First, I believe that nobody believes that >1000 proteins really interact with the kinases and are likely to be mostly non-specific precipitating proteins.

Second, it does not appear that a proper control was used to subtract these non-specific proteins from the list (i.e. a blank IP from a non-transformed line). As the end result of the manuscript does not rely on this data other than as an entry point to understand how these kinases were selected for further study, I do not think the list of proteins (or the quantitation) is a major issue. However, lacking the proper controls, the authors should avoid referring to this data as revealing “SnRK2-interacting proteins” as this is quite misleading and not supported by the presented data.

>>>ANSWER 10

In response to this comment, we have revised the description of the protocol used for co-IP coupled with LC-MS/MS analysis (L105-106). Moreover, we performed the same experiments using GFP-expressing plants as the appropriate control and excluded the proteins that were detected in the plants from the list of candidate proteins (L107-110). We also provide an explanation for our focus on protein kinases and phosphatases among the candidate proteins (L110-115).

>>> COMMENT 11

2. In line 188 when referring to MBP-fused VCS fragment in the in-gel kinase: I think it is asking quite a bit of a casual reader to remember from the brief mention in the Introduction that Varicose (VCS) was previously shown to be phosphorylated by subclass I SnRK2s. Bringing in that information to inform the experimental design would be extremely helpful.

>>>ANSWER 11

According to this suggestion, we have revised the sentence to indicate that VCS was previously shown to be phosphorylated by subclass I SnRK2s (L196-197).

>>> COMMENT 12

3. Although it isn't necessary for this paper for the authors to know the identity of the higher MW non-specific kinase activity in Fig 2d (and following), I don't think that it can go without mentioning that there is another dehydration-related kinase activity present that is OSRK1/2/3-independent. It is so obvious that it needs to be at least mentioned.

>>>ANSWER 12

We believe that the higher MW kinase activity in Fig. 2d indicates an unknown VCS-phosphorylating kinase(s) that is likely a mitogen-activated protein (MAP) kinase(s) based on the molecular weight. In response to this comment, we have marked the band with an asterisk and provide an explanation in the legend for Figure 2. We also provide a description of this kinase activity in the Results section (L234-237).

>>> COMMENT 13

4. Line 247: I believe the authors meant to say “Similar to the srk2abgh plants” rather than “In contrast to ...”

>>>ANSWER 13

According to this helpful comment, COMMENT 7 provided by Reviewer #1 and COMMENT 30 provided by Reviewer #3, we have revised the sentence to ensure accuracy (L283-286).

>>> COMMENT 14

5. Lines 258-261: This sentence needs to be rewritten as there are a number of errors in both construction and spelling.

>>>ANSWER 14

According to this helpful suggestion, we have corrected this sentence (L295-300).

Reviewer #3 (Remarks to the Author):

>>> COMMENT 14

The article by Soma et al. titled Plant Raf-like kinases regulate the mRNA population upstream of ABA unresponsive SnRK2 kinases under drought stress describes the role of subclass I SnRK2s in osmotic stress in particular their interaction and potential phosphorylation by three B4 Raf like MAPKKKs. They employed biochemical analysis and phenotypic assessments on mutants to derive the functional significance of the activation via phosphorylation of subclass I SnRK2s in plant growth under osmotic stress. Overall, their work highlights an important novel contribution on the role of subclass SnRK2s and the Raf like MAPKKKs in osmotic stress and plant growth promotion, which is of interest in wider field in plant biology and biochemistry.

Here I highlight comments and concerns towards improving this work for publication in particular the need for further evidence on target phosphorylation sites that are only targeted by OSRKs.

>>>ANSWER 15

Thank you very much for your kind comments. We have revised our manuscript according to your valuable comments as described below.

>>> COMMENT 16

Major revisions

With the given data in particular supplementary data on IP, it is highly unlikely that an

independent research can reproduce the work without the data evidence from this research. This data needs to be provided.

>>>ANSWER 16

In response to this comment, we have shown the data in the new Supplementary Table 1.

>>> COMMENT 17

Results

Line 105: By two independent LC-MS/MS runs, do the authors imply technical replicates or biological replicates. It is not clear. If it is technical replicates, I suggest the authors perform biological replicates in addition.

In addition, there is no data or mention about important co-IP controls such as co-IP from a control GFP expressing plant line or just GFP IP on the wild type plants and how they identify false positive? How did the authors quantify the candidates co-IP proteins, in particular those that the authors selected. Were they expressed quantitatively in the same manner between the SRK2A and SnRK2G.

>>>ANSWER 17

In response to this comment and COMMENT 10 provided by Reviewer #2, we have revised the description of the protocol used for co-IP coupled with LC-MS/MS analysis (L105-106). Moreover, we performed the same experiments using GFP-expressing plants as an appropriate control and excluded the proteins that were detected in the plants from the set of candidate proteins (L107-110). We also provide a description for our focus on protein kinases and phosphatases among the candidate proteins (L110-115).

We previously reported that the SRK2A-GFP and SnRK2G-GFP protein levels were similar in the transgenic plants (Soma et al., Nature Plants **3**, 16204 (2017)) and indicate this result in the Methods section (L484-486).

>>> COMMENT 18

Line 124: For the luciferase assay, did the authors perform this experiment with SRK2G? If not how could they conclude that OSRK1-3 are potent candidate novel interactors of SnRK2s.

>>>ANSWER 18

In response to this comment, we performed the experiment using SRK2G-cLUC. The results are shown in Supplementary Figure S1 and described in the Results section (L131-134).

>>> COMMENT 19

Paragraph starting on line 165: Although the data suggest that the three OSRKs positively regulate the activity of the subclass I SnRK2s under osmotic stress, the authors show an indirect potential that OSRKs may phosphorylate SnRK2s and that this phosphorylation is effective for SnRK to catalyse VCS compared to the autophosphorylated form of SnRK2, I

think it is important to determine the exact target residues for OSRK phosphorylation that are not subject to autophosphorylation. As it stands, it leaves a hollow in the study that greatly adds value to the role of OSRK in interacting and phosphorylating SRK2 - if its not just facilitating the autophosphorylation of SnRK2s. After the identification of the phosphosites by mass spectrometry authors should correlate with the bioinformatic predictions to unequivocally prove that the sites obtained are indeed targets of OSRKs.

>>>ANSWER 19

In response to this comment, we identified the phosphorylation sites of SRK2G by mass spectrometry and show these sites in Supplementary Table 2. We found that Ser-154 in SRK2G is one of the direct phosphorylation targets of OSRKs (Supplementary Fig.6). We provide an explanation of and discuss the results in the Results section (L205-220).

>>> COMMENT 20

Line 319: Is this statement supported by data: Thus, we concluded that the three OSRKs are quickly activated by phosphorylation?

>>>ANSWER 20

In response to this comment, we revised this sentence as follows (L360-363): “ Thus, this study highlights the existence of three OSRKs that activate downstream subclass I SnRK2s in response to osmotic stress to form the ABA-independent OSRK-subclass I SnRK2-VCS signalling cascade (Fig. 4g).”

>>> COMMENT 21

Minor revisions

Define terms like ARK, EIN2 and ETR1 and the rest of the abbreviations not defined?

Check your abbreviations and make sure they are used at least once. For those not, could you remove them

>>>ANSWER 21

We have defined the abbreviations that have not been defined, including ARK, EIN2 and ETR1. We also checked all the abbreviations throughout this manuscript.

>>> COMMENT 21

Number of biological and/or technical replicates used need to be stated for example for the co-IP and for transcriptomics.

>>>ANSWER 22

In response to this comment, we have shown the number of biological replicates used for co-IP coupled with LC-MS/MS (L105-106) in the Results section and those used for RNA

sequencing (L592-593 and 598-599) and qRT-PCR (L592-593 and 610-611) in the Methods section.

>>> COMMENT 23

Supplementary table 1 should also contain all the co-IP proteins identified showing which ones are detected in both replicates and which ones are common between the SRK2A and 2G-GFPs

>>> ANSWER 23

In response to this comment, we have shown all the identified co-IP proteins and indicate which ones were detected in both replicates and which ones were common to the SRK2A and 2G-GFPs in the new Supplementary Table 1.

>>> COMMENT 24

1. Line 27: change 'osmotic stress to abiotic stress. This sentence is a generalized statement.

>>> ANSWER 24

Because SnRK2s have been shown to play important roles, particularly in the response to osmotic stress, we would like to use osmotic stress in this sentence.

>>> COMMENT 25

2. Line 40: Rephrase as follows- Abiotic stresses such as drought and high salinity and consequently osmotic stress are

>>> ANSWER 25

In response to this comment, we have revised this sentence (L40-41).

>>> COMMENT 26

Line 101-102: Are these over expressing lines? Its not clear in both the methods and the results sections.

>>> ANSWER 26

We have revised the sentence describing these plants in the Methods section (L484-486).

>>> COMMENT 27

Line 118-119: Did the authors perform the same experiment with OSRK2 and 3? Its not clear.

>>> ANSWER 27

We performed this experiment using OSRK1 as the representative of the three OSRKs. In response to this comment, we have revised the sentence (L124-125).

>>> COMMENT 28

Authors need to clearly define their terms, they are using dehydration and osmotic stress

interchangeably and in some cases for example Line 240 which is heading for water-deficit experiment, they should clearly explain the link to osmotic stress bearing in mind that one is upstream of the other.

>>>ANSWER 28

We used “dehydration” only when referring to the dehydration treatment, and “osmotic stress” was used as a general term for dehydration, high salinity, and/or mannitol treatment. We have also changed “water deficit” to “drought” or “osmotic stress” as appropriate.

>>> COMMENT 29

Line 242: under osmotic stress conditions. This could rather be rephrased to..... under osmotic stress inducing conditions.

>>>ANSWER 29

We have rephrased the indicated phrase to “under osmotic stress inducing conditions” (L278).

>>> COMMENT 30

Line 247: Can the authors clearly explain what they mean by poorer growth- how does one measure poorer growth? I suggest that the authors explain by means of phenotypic traits

>>>ANSWER 30

In response to this comment, COMMENT 7 provided Reviewer #1 and COMMENT 13 provided by Reviewer #2, we have revised this sentence (L283-288).

>>> COMMENT31

Line 248: term drought stress- is this for water deficit? Authors need to maintain wording particularly when referring to stresses or at least indicate that x and y are the same.

>>>ANSWER 31

We decided to use “drought stress” instead of “water deficit” as described in ANSWER 25.

>>> COMMENT 32

Line 249: by using the term "expressing" do the authors mean complementation

>>>ANSWER 32

According to this comment, we revised this sentence (L286-288).

>>> COMMENT 33

Line 260: replace poorer with reduced or retarded

>>>ANSWER 33

We have replaced “poorer” with “reduced” or “retarded” (L293 and 298)

>>> COMMENT 34

Line 275: do the authors mean osmotic stress?

>>>ANSWER 34

We mean dehydration stress (L315).

>>> COMMENT 35

Lline 276: deleted showed and replace reduced by decreased in

>>>ANSWER 35

We have revised the indicated sentence according to this suggestion (L316-317).

>>> COMMENT 36

Line 280: Now it seems like the authors did a drought stress treatment for srk2abgh plants and dehydration for the osrk1/2/3 plants. Correct all your terminology in the paper and be consistent.

>>>ANSWER 36

We have changed “drought” to “dehydration” (L320, 329 and 332).

>>> COMMENT 37

Line 298: Do the authors indicate that SnRK2s are functionally activated in their kinetic role or they are referring to gene induction and functional activation? its not quite clear!

>>>ANSWER 37

We removed “quickly” and revised this sentence to clarify this point (L339-341).

>>> COMMENT 38

Line 301- replace quickly with promptly

>>>ANSWER 38

We have replaced “quickly” with “promptly” (L342).

>>> COMMENT 39

Line 303: Replace As by Since

>>>ANSWER 39

We have replaced “As” with “Since” (L344).

>>> COMMENT 40

Line 305-306: Although the authors indicate their hope of elucidating the specific subclass 1 SnRK2s interactors, did they attempt to measure the level of ABA at this 10 mins time point?

>>>ANSWER 40

Because it is believed that subclass I SnRK2s are activated within 10 min, we selected this condition for the screening of subclass 1 SnRK2-interacting proteins by co-IP coupled with

LC-MS/MS analysis. We believe that the level of ABA is not as important because subclass 1 SnRK2s are not activated by ABA.

>>> COMMENT 41

Line 308: insert "which were " before among...

>>>ANSWER 41

We have inserted “which were ” before “among” (L349).

>>> COMMENT 42

Line 309: replace 'of the previous' by from the previous

>>>ANSWER 42

We have replaced “of the previous” with “from the previous” (L350).

>>> COMMENT 43

Line 309-311: rephrase the sentence

>>>ANSWER 43

We have revised this sentence (L350-353).

>>> COMMENT 44

Line 324: correct MAPKKs to MAPKKKs

>>>ANSWER 44

We have corrected MAPKKs to MAPKKKs (L366).

>>> COMMENT 45

Line 330: By separation do the authors mean evolutionary divergent or something else. This needs clarification.

>>>ANSWER 45

In response to this comment, we have revised this sentence (L373-375).

>>> COMMENT 46

Line 335: replace 'to be' with 'to have been'

>>>ANSWER 46

We have replaced “to be” with “to have been” (L379).

>>> COMMENT 47

Line 336: delete 'levels'

>>>ANSWER 47

We have deleted “levels” (L380).

>>> COMMENT 48

Line 337: replace 'but not activated by ABA' either by ' but not ABA' or 'and not ABA'

>>>ANSWER 48

We have replaced “but not activated by ABA” by “but not ABA” (L381).

>>> COMMENT 49

Line 351: Six members of which subclass of SnRK2s- be clear

>>>ANSWER 49

We have revised the sentence and sited Supplementary Fig. S2 to clarify these six members of the B3 subgroup (L396-397).

>>> COMMENT 50

Line 373: replace 'the gene expression that was upregulated and downregulated in these....' by 'the differentially regulated genes in these.... '

>>>ANSWER 50

We have replaced “the gene expression that was upregulated and downregulated in these” with “the differentially regulated genes in these” (L419).

>>> COMMENT 51

Line 378: delete ‘the’

>>>ANSWER 51

We have deleted the term “the” in this sentence (L423).

>>> COMMENT 52

Line 403: State the composition of the germination medium

>>>ANSWER52

We have stated the composition of the germination medium (GM) in the Methods section (L451-454).

>>> COMMENT53

Line 434: Replace "... were previously used..." by "were used as previously described"

>>>ANSWER 53

We have replaced “were previously used” with “were used as previously described” (L485).

>>> COMMENT 54

Line 423: Can the authors include a brief description of the immunoblot analysis technique used.

>>>ANSWER 54

In response to this comment, we have included a description of the immunoblot analysis in the Methods section (L543-552).

>>> COMMENT 55

Line 448: Add 'of' in the sentence The mobile phase....

>>>ANSWER 55

We have added the term “of” to the sentence (L499-500).

>>> COMMENT 56

Line 450: the A:B = 60:40 in 300 min, Is this correct? How long is the overall run time for each sample?

>>>ANSWER 56

This is correct. The overall run time is approximately 6.5 h for each sample. We analysed more than 30 samples, which took approximately 12 days.

>>> COMMENT 57

Line 510-12: The sentence is not clear on how the plants were grown, for how long each in the GM agar and in the soil prior to water withdrawal.

>>>ANSWER 57

According to this comment, we have corrected this sentence (L576-578).

>>> COMMENT 58

Figures:

1. state the number of reps in Figure 1

>>>ANSWER 58

We have stated the number of repetitions in Figure 1.

>>> COMMENT 59

2. Figure 1b does not show the IP controls e.g. GFP or mCherry IP for background. This should also be explained in the co-IP and mass spectrometry data analysis in the methods and results sections

>>>ANSWER 59

We performed the co-IP experiment using an anti-mCherry antibody and mCherry IP as the IP control and replaced the previous Fig. 1b with the new results including those obtained with mCherry IP as the background.

>>> COMMENT 60

3. Figure d-f correct spelling for merged from marged

>>>ANSWER 60

We have corrected the spelling of “marged” to “merging” in Figure 1d-f.

REVIEWERS' COMMENTS:

Reviewer #1 (Remarks to the Author):

The authors have appropriately addressed my concerns. I support publication of the paper in its current form.

Reviewer #3 (Remarks to the Author):

The revised article by Soma et al. on the Plant Raf-like kinases regulate the mRNA population upstream of ABA unresponsive SnRK2 kinases under drought stress describes the role of subclass I SnRK2s in osmotic stress in particular their interaction and potential phosphorylation by three B4 Raf like MAPKKKs. The revised version of the article reads well and clearly addressed all the concerns raised. In addition, the authors provided additional data to support their claims and further to strengthen their arguments.

A few minor changes are needed:

- Spelling errors for example: Line 577 - plats for plates
- Line 500 Add "of" before 2% in the sentence "...mobile phase comprised 2% ..."

Response to Reviewers

REVIEWERS' COMMENTS:

Reviewer #1 (Remarks to the Author):

>>>COMMENT I

The authors have appropriately addressed my concerns. I support publication of the paper in its current form.

>>>ANSWER I

Thank you very much for your comment. We appreciate your support.

Reviewer #3 (Remarks to the Author):

The revised article by Soma et al. on the Plant Raf-like kinases regulate the mRNA population upstream of ABA unresponsive SnRK2 kinases under drought stress describes the role of subclass I SnRK2s in osmotic stress in particular their interaction and potential phosphorylation by three B4 Raf like MAPKKKs. The revised version of the article reads well and clearly addressed all the concerns raised. In addition, the authors provided additional data to support their claims and further to strengthen their arguments.

>>>COMMENT I I

A few minor changes are needed:

- Spelling errors for example: Line 577 - plats for plates
- Line 500 Add "of" before 2% in the sentence "...mobile phase comprised 2% ..."

>>>ANSWER I I

Thank you very much for pointing out these two mistakes. We have corrected them.